# Transmission of tauopathy strains is independent of their isoform composition

Zhuohao He [1,3]*, Jennifer D. McBride[1], Hong Xu[1], Lakshmi Changolkar[1], Soo-jung Kim[1], Bin Zhang[1], Sneha Narasimhan[1], Garrett S. Gibbons [1], Jing L. Guo[1], Michael Kozak[1], Gerard D. Schellenberg[2], John Q. Trojanowski [1] & Virginia M.-Y. Lee[1]*

The deposition of pathological tau is a common feature in several neurodegenerative tauopathies. Although equal ratios of tau isoforms with 3 (3R) and 4 (4R) microtubule-binding repeats are expressed in the adult human brain, the pathological tau from different tauopathies have distinct isoform compositions and cell type specificities. The underlying mechanisms of tauopathies are unknown, partially due to the lack of proper models. Here, we generate a new transgenic mouse line expressing equal ratios of 3R and 4R human tau isoforms (6hTau mice). Intracerebral injections of distinct human tauopathy brain-derived tau strains into 6hTau mice recapitulate the deposition of pathological tau with distinct tau isoform compositions and cell type specificities as in human tauopathies. Moreover, through in vivo propagation of these tau strains among different mouse lines, we demonstrate that the transmission of distinct tau strains is independent of strain isoform compositions, but instead intrinsic to unique pathological conformations.

[1] Department of Pathology and Laboratory Medicine, Institute on Aging and Center for Neurodegenerative Disease Research, University of Pennsylvania School of Medicine, Philadelphia, PA 19104, USA. [2] Department of Pathology and Laboratory Medicine, Penn Neurodegeneration Genomics Center, University of Pennsylvania School of Medicine, Philadelphia, PA 19104, USA. [3] Present address: Interdisciplinary Research Center on Biology and Chemistry, Shanghai Institute of Organic Chemistry, Chinese Academy of Sciences, 201210 Shanghai, China. *email: hezh@sioc.ac.cn; vmylee@upenn.edu

Neurodegenerative tauopathies, including Alzheimer's disease (AD), corticobasal degeneration (CBD), progressive supranuclear palsy (PSP), and Pick's disease (PiD), are major neurodegenerative diseases characterized by the deposition of aggregated tau proteins in neurons, as well as glia in CBD, PSP, and PiD. Although tau is the disease protein found as cellular inclusions, patients with distinct tauopathies show different clinical symptoms and patterns of tau aggregates[1]. Importantly, analyses of postmortem tauopathy brains showed that the accumulation and stereotypic spread of tau pathology over time plays important roles in the progression of these diseases, as the clinical symptoms correlate with the dysfunctions of different brain regions that harbor tau inclusions in tauopathy patients[2,3]. Thus, understanding the underlying mechanisms of pathogenesis in different tauopathies, is critical for developing diagnostics and interventions for these disorders.

Tau proteins are encoded by the *MAPT* tau gene, and full length tau has two N-terminal domains and four microtubule-binding repeat domains. In adult human brains, alternative RNA splicing of exons 3 and 10, encoding the 2nd N-terminal domains and the 2nd microtubule-binding repeat domain, respectively, result in the expression of six tau isoforms, with an equal ratio of the isoforms containing 3 (3R) or 4 (4R) microtubule-binding repeat domains. Tau expression is developmentally regulated, such that in adult human brain, all six tau isoforms are expressed, while in fetal brain, only shortest 3R tau isoform is expressed[4]. Although predominantly expressed in neurons, tau expression has also been reported in cultured oligodendrocyte[5,6], but expression in other glial cells such as astrocyte and microglia is unclear. Within neurons, the different tau isoforms were reported to have different subcellular distributions[7,8]. In different brain regions, tau isoform expression pattern is also differentially regulated[9–11]. Distinct tau isoforms have also been reported to have different functions[12].

Each tauopathy has a unique pattern of neuropathology, rate of progression, and cellular and regional involvement. As a result, tau inclusions from diverse tauopathy lesions in brains with different properties are considered as distinct strains[1,2]. In AD, pathological tau aggregates known as neurofibrillary tangles (NFTs) comprises paired helical filaments assembled from all six tau isoforms in neurons, whereas in CBD and PSP, tau-positive inclusions consist predominantly of 4R tau are found in neurons, oligodendrocytes and astrocytes. In contrast, PiD is characterized by Pick bodies in neurons, as well as tau aggregates in glia, which are composed predominantly of 3R tau isoforms. It is unclear how such distinct tau isoform compositions in the strains correlate with their unique pathogenic properties. A major obstacle to address this question is the lack of informative animal models with tau expression pattern similar to human with six isoforms and an equal 3R and 4R ratio.

Furthermore, recent reports have implicated a unique self-propagating mechanism to explain the progression or spread of tau pathology, that pathological tau protein could transmit their pathological conformations to the physiological tau protein, converting tau protein from normal form into pathological form. We recently developed sporadic tauopathy models that recapitulated the transmission of distinct tau strains in wild-type (WT) mice[13,14], but it is still unclear why tauopathies comprises distinct tau isoforms, a key feature of tau strains, since adult WT mouse brain only express 4R tau isoforms. To elucidate the transmission properties of tau strains, we inoculated different tau aggregates from distinct human tauopathy brains into a newly developed human tau transgenic (Tg) mouse line expressing equal ratios of 3R and 4R human tau (Htau) in the brain without endogenous mouse tau (6hTau). Using this novel model, we explored how distinct tau isoform compositions affect strain transmission properties. Here, we show distinct tau strain transmission pattern is independent of its isoform compositions.

## Results

### Generation of 6hTau mice with equal 3R and 4R tau isoforms.

To study the pathogenesis of different tau strains, we first generated a new Tg mouse line (designated as 6hTau mice) expressing both 3R and 4R Htau isoforms in a 1:1 ratio similar as in human brains (Fig. 1a–d). To generate the 6hTau mice, we first crossed the previously described hT-PAC-N mouse line[7] to a mouse *MAPT* knockout (KO) line, resulting in the expression of all six WT Htau isoforms but with much higher 3R than 4R tau isoforms. We then bred these mice with another mouse line (E10 + 14) that carried the human *MAPT* gene harboring a mutation in the intron near exon 10 (E10 + 14) and expressed higher levels of 4R than 3R WT Htau. The generated 6hTau mice only express six WT Htau isoforms, with a 3R to 4R tau ratio of ~1. The 0N3R and 0N4R human tau isoforms were most abundant in 6hTau mice, which differs slightly from human brains wherein the 1N3R and 1N4R tau isoforms predominate, might be due to the different RNA splicing mechanism between mouse and human (Fig. 1c, d). The total tau expression level in 6hTau mice is about twice as much as endogenous mouse tau (Mtau) in WT mice, and three folds as much as Htau in normal human brains (Fig. 1b). Similar to WT mice, the levels of total tau and the pattern of 3R and/or 4R expression did not change after 2 months of age and remained constant throughout the lifespan of the mice (Fig. 1e–i). Regional analyses of tau expression by western blots showed that, similar to WT mice, total tau levels in 6hTau were higher in hippocampus and cortex compared to other brain regions. Furthermore, relative total tau express in cerebellum and spinal cord were relatively higher in 6hTau mice when compared to the counterparts in human brains (Fig. 1j, k). Finally, the expression pattern of the six isoforms among different CNS regions of 6hTau mice were similar, except in the spinal cord, where the 110 KDa big tau isoform is also expressed, and the 1N isoforms, instead of the 0N isoforms in other regions, are predominant, same as human spinal cord tau expression (Fig. 1l–n). It is notable that the 6hTau mice did not spontaneously develop tau pathologies up to 25 months of age (Supplementary Fig. 1a). Even though the intracerebral injections of preformed fibrils (PFFs) made from recombinant 3R, 4R or mixture of 3R and 4R tau proteins, could induce tau pathologies in Tg mice-expressing human mutant tau (PS19), but not in non-Tg WT mice[13,15], similar injections of recombinant tau PFFs in 6hTau mice did not induce any tau pathologies (Supplementary Fig. 1b, c).

### Tau strains show diverse tau isoform compositions.

To induce pathologies in 6hTau mice, we used pathological tau extracted from human tauopathy brains as "seeds"[13,14,16] for injection into 6hTau mice. As expected, tau extracted from these different human tauopathy brains showed that AD-tau proteins were comprised of both 3R and 4R tau, PiD-tau proteins were predominantly 3R tau, and CBD-, PSP-tau proteins were predominantly 4R tau (Supplementary Fig. 2).

### Tau strains induce isoform-matched pathologies in 6hTau mice.

To determine if distinct tau strains from diverse human tauopathies studied here can be recapitulated in mice, we propagated the tau strains in vivo by intracerebrally injecting 1 μg/site of different human brain-derived tau strains (designated as AD_P0-, PiD_P0-, CBD_P0-, PSP_P0-tau) into the hippocampus and overlying cortex of 6hTau mice. Hereafter, the specific human tau strain is denoted first and P0 refers to the initial human brain-derived tau pathology. Thus, the injected mice are

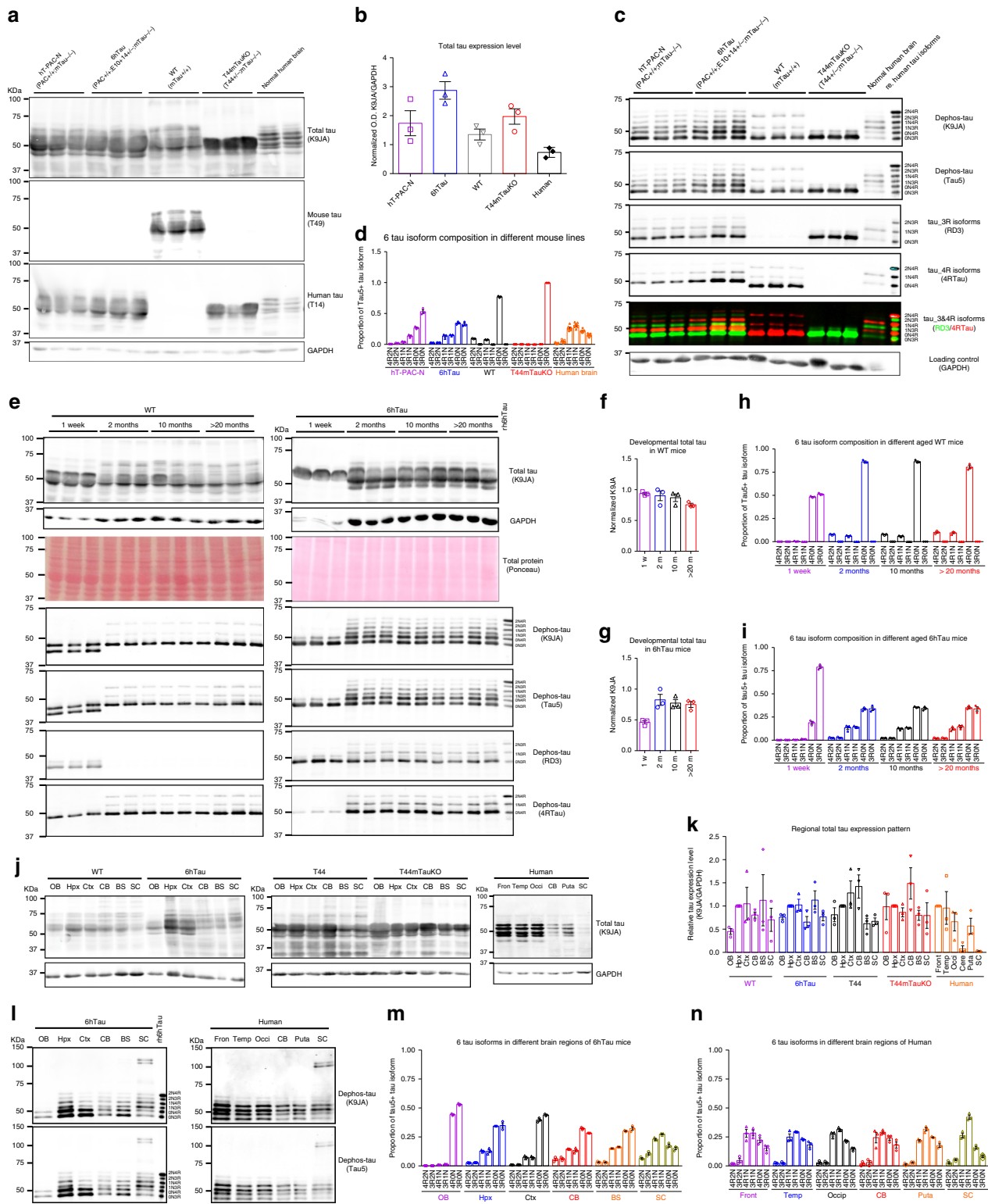

designated hereafter as AD_P0-6hTau, PiD_P0-6hTau, CBD_P0-6hTau and PSP_P0-6hTau, respectively (Fig. 2a, b). Immunohistochemistry (IHC) using monoclonal antibody (Mab) AT8 to hyperphosphorylated tau showed that all the human tau strains induced significant amounts of tau pathologies in the hippocampus at 3 months post injection (m.p.i.) (Fig. 2b and Supplementary Fig. 3a), while the control brain lysate extracted in a similar way did not induce significant tau pathologies in the 6hTau mice (Fig. 2c). The pathological tau induced in these

6hTau mice were also hyperphosphorylated at Ser396, Ser404 (PHF-1), Thr231 (AT180), Ser262 (12E8), and misfolded (MC1), as well as positive for Gallyas silver staining except the PiD-tau induced tau pathologies, consistent with previous report (Supplementary Fig. 4)[17]. IHC using 3R- or 4R-specific anti-tau antibodies showed the induced tau pathologies in 6hTau mice recapitulated the isoform compositions in the original human tau strains since injection of AD-tau from two AD cases induced tau pathologies comprised of both 3R and 4R tau, while tau extracts

**Fig. 1 Characterization of the new 6hTau and T44mTauKO Tg mice. a** Comparison of total tau expression levels among the hT-PAC-N, 6hTau, WT, and T44mTauKO mouse lines, as well as human brain tau. Homogenates from cortices of the different mouse lines at the age of around 3 months or from normal human brains were probed with pan tau antibody K9JA, Mtau selective antibody T49, and Htau selective antibody T14. GAPDH served as a loading control. **b** Quantification of the relative total tau expression levels among the different brains as shown in **a**, n = 3 for each group. **c** Tau isoform expression pattern among brains of the different lines. Tau in brain homogenates were dephosphorylated with lambda protein phosphatase and probed, respectively, with antibodies K9JA and Tau5, 4R isoform tau-specific polyclonal antibody 4R Tau and 3R isoform tau-specific Mab RD3. The recombinant (re) six human tau isoforms were, respectively, loaded as standards on the right lanes. 6hTau mouse line express only human six tau isoforms, while T44mTauKO mouse line express only the shortest human 3R tau isoform, 0N3R. Loading control GAPDH were probed using same amounts of lysates without dephosphorylation. **d** Quantification of different tau isoform expression pattern in each mouse line. **e** developmental expression pattern of total tau and isoforms in WT and 6hTau mice at indicated ages. rh6hTau, recombinant six human tau isoforms. Quantification of the relative total tau expression levels in **f**, WT and **g**, 6hTau mice. Different isoform expression patterns during developmental stages in **h** WT and **i** 6hTau mice. **j** Blot images and **k** Quantifications for brain regional expression patterns of total tau among WT, 6hTau, T44, T44mTauKO mouse lines, and normal human brains were probed by tau antibody K9JA. GAPDH served as a loading control. **l** immunoblot images and quantifications of tau isoform expression patterns in different brain regions in **m** adult 6hTau mice and **n** normal human controls. OB olfactory bulb, Hpx hippocampus, Ctx cortex, CB cerebellum, BS brainstem, SC spinal cord, Fron frontal cortex, Temp temporal cortex, Occi occipital cortex, Puta putamen. Quantifications are presented as mean ± s.e.m., with each dot representing an individual. Source data is available as a Source Data file.

from two PiD cases predominantly induced 3R tau pathologies. In contrast, tau extracts from both CBD and PSP case predominantly induced 4R tau pathologies at 3 m.p.i. (Fig. 2b, h and Supplementary Fig. 3a). Since cases extracted from each type of tauopathy brain behaved similarly in recruiting corresponding tau isoforms during the in vivo seeding processes (Fig. 2b and Supplementary Fig. 3a), we mainly used one case of each tau strain to simplify the study, unless specified. The different isoform recruitment pattern by distinct human tau strains remained consistent up to 6 m.p.i. (Supplementary Fig. 3b), suggesting that distinct tauopathy strain isoform recruitments are recapitulated faithfully in the 6hTau mice.

**Isoform-specific seeding is determined by intrinsic property.** To test if the isoform-specific transmission induced by distinct tau strains is due to pathological tau themselves or due to contaminating co-injected human brain materials (as the seeded materials contained <30% pathological tau protein), we performed serial in vivo propagation of the induced tau pathologies by repetitive in vivo seeding (AD_P1, PSP_P1, CBD_P1, PiD_P1, AD_P2, PSP_P2, etc Fig. 2a). If the tau strain properties are dependent on the contaminating human proteins in the injected materials, then such tau-isoform-specific recruitment would not be faithfully maintained during the serial in vivo propagation in the 6hTau mice. However, injection of sarkosyl-insoluble extracts from the hippocampus of each of AD_, PiD_, PSP_, and CBD_P0-6hTau mice into naive 6hTau mice (Fig. 2a) showed that the induced tau pathology by AD_P1 still contained both 3R and 4R tau, while PiD_P1 predominantly recruits 3R tau. In contrast, PSP_P1 and CBD_P1 showed only 4R tau recruitment, while the control lysate Ctrl_P1 still did not induce any tau pathologies (Fig. 2c, d, i). Continued repetitive propagation in vivo with additional 6hTau mice showed that AD_P2 still recruited both 3R and 4R tau, while PSP_P2 and CBD_P2 only recruited 4R tau. The PiD-tau was less efficient in inducing tau pathologies compared with the other tau strains so it was more challenging to propagate PiD_3R tau continuously in vivo (Fig. 2e, j). Owing to the limited yield of strain_P1, P2 from the injected mice, we could not inject as much propagated pathological tau as P0 into the 6hTau mice in this in vivo propagation experiment, resulting in lower pathologies induced by strain_P1 and strain_P2 when compared with P0. Altogether, these serial in vivo propagation results suggest the isoform-specific transmission by distinct tau strains is more likely due to pathological tau themselves rather than other co-factors.

To further confirm the isoform recruitment, we conducted immunoblot analyses on the sarkosyl-insoluble tau fractions from

injected 6hTau mice, and probed them with 4R- or 3R- specific anti-tau antibodies. These results were consistent with the IHC data that induced tau pathologies in AD_P1 and P2 were detected by both 3R and 4R tau antibodies, whereas PSP_P1 and P2 as well as CBD_P1 and P2 were mainly detected by 4R tau antibody. Meanwhile, the PiD_P1 was mainly detected by 3R tau antibody (Fig. 2f, g), suggesting that distinct tauopathy strains prefer to recruit the corresponding tau isoforms seen in AD, PSP, CBD, and PiD into the corresponding tau aggregates in the 6hTau mice, confirming this process is dependent upon the tau strain itself.

**Cross-seeding non-corresponding tau isoforms is inefficient.** The above data showed that different tau strains preferred to recruit corresponding tau isoforms in the presence of both 3R and 4R tau monomers during in vivo seeding process, but it was unclear whether the tau strains were also able to cross-recruit non-corresponding tau isoforms in the absence of corresponding isoform tau expression. For example, we asked whether 3R tau strain PiD-tau is capable of recruiting 4R tau monomers in mice expressing only 4R tau, or if 4R tau strains PSP-tau and CBD-tau could also recruit 3R tau monomers in mice expressing only 3R tau? To address these questions, we used adult WT mice as a 4R tau-only expressing mouse line, and generated a 3R-tau only expressing mouse line T44mTauKO by crossing human 3R tau isoform expressing T44 Tg mice[18] with mouse *MAPT* gene KO mice (Fig. 1a–d). Similar to the original T44 mice, this new mouse line showed similar total regional tau expression levels (Fig. 1j, k), and by postnatal 8 months of age, they merely developed granule-like tau pathologies in the brainstem, but rarely develop neurofibrillary tangle-like tau inclusions in other brain regions (Supplementary Fig. 1d). We, respectively, injected distinct strain_P0 into either 4R Tau mice (WT) or 3R Tau mice (T44mTauKO), and detected the induced tau pathologies using AT8 Mab (Fig. 3a). While control brain lysates did not induce significant tau pathologies in either 4R Tau or 3R Tau-expressing mice (Fig. 3b), AD_P0-tau, PSP_P0-tau, and CBD_P0-tau, which all contain 4R isoform tau, could induce pathological tau in 4R Tau mice. However, same amount of PiD_P0-tau, which is predominantly comprised of 3R tau induced negligible or no tau pathologies in the 4R Tau mice up to 6 m.p.i. (Fig. 3c, e, f), suggesting 3R tau strains are very inefficient in recruiting 4R tau to form pathological tau aggregates. On the other hand, both AD_P0-tau and PiD_P0-tau, which contain 3R tau isoforms, induced robust pathological tau in 3R Tau mice. However, similar amount of PSP-tau and CBD-tau, which are predominantly comprised of 4R tau isoforms, induced negligible tau pathologies in 3R Tau Tg mice even up to 6 m.p.i. (Fig. 3d, e, f), suggesting

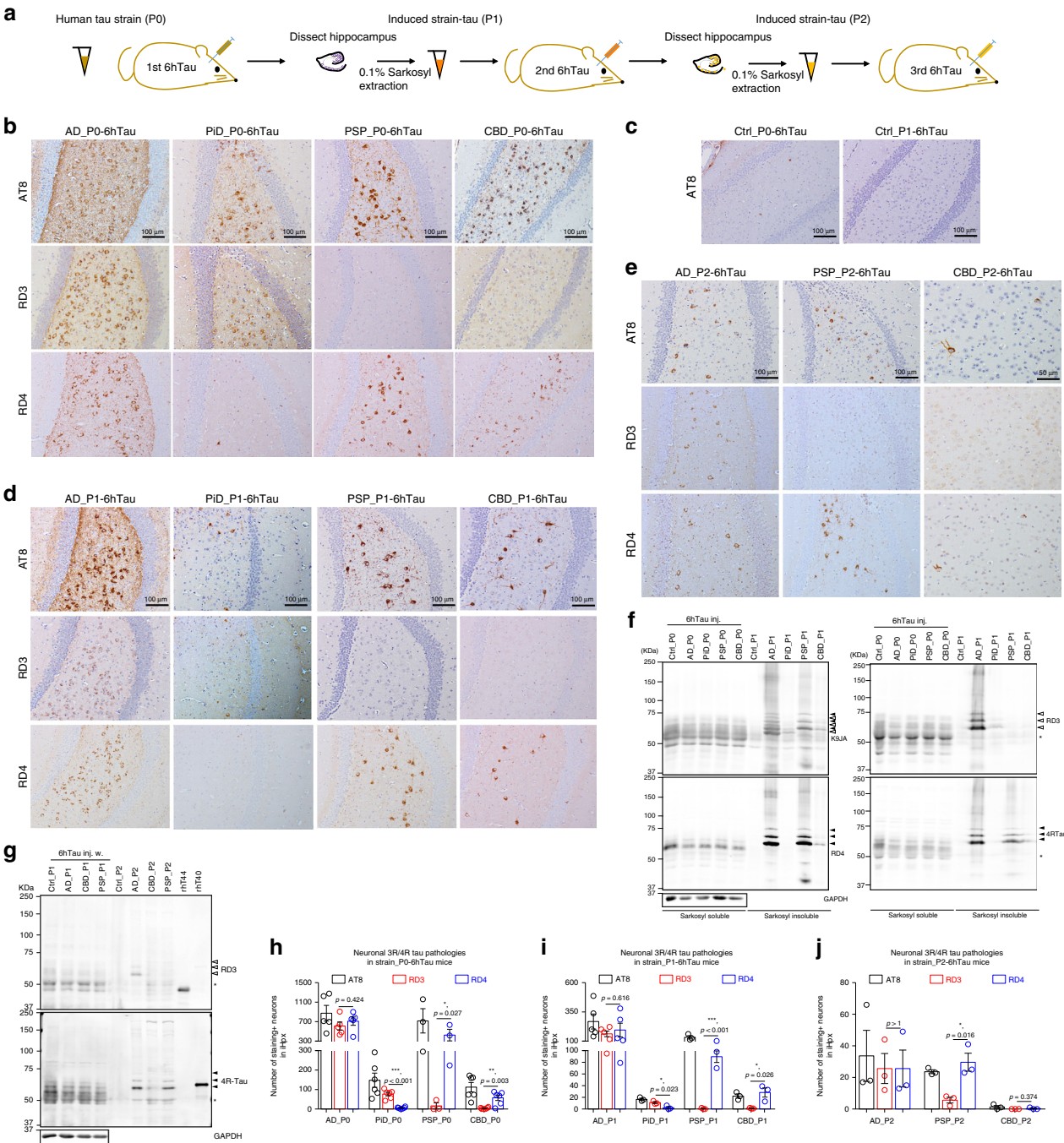

**Fig. 2 Isoform-specific seeding pattern is unaltered during repeated propagation. a** Schematic showing in vivo propagation paradigm in 6hTau mice. Similar amounts of human tau strains (P0) were injected into the hippocampus of naive 6hTau mice for 1st round transmission and 3 months later, the hippocampi were dissected and extracted using detergent to obtain the induced tau strain (P1). The P1 tau strain was re-injected into the hippocampus of naive 2nd round transmission 6hTau mice, and the induced P2 tau pathologies were extracted similarly as P1 and injected into the hippocampus of 3rd round of 6hTau mice. Representative IHC staining in panel **b** using MAbs AT8, MAb RD3 (3R isoform tau-selective) and MAb RD4 (4R tau isoform-specific) on adjacent brain sections from 6hTau mice injected with equal amounts of P0-tau from different human tau strains; or **c** with control lysate from normal human brains Ctrl_P0 or non-injected aged 6hTau mouse brains Ctrl_P1; or **d** with induced P1-tau; or **e** with induced P2-tau at 3 m.p.i. The P1 injection doses were 0.1–0.5 μg per site, and P2 were 0.01–0.03 μg per site, while the P0 were 1 μg per site. Biochemical extraction of induced **f** P1 from strain_P0–6hTau and **g** P2 from strain_P1–6hTau mice with 0.1% sarkosyl. The sarkosyl-insoluble tau were probed with Mab RD3, Mab RD4, and polyclonal anti-4R Tau-specific antibody to examine the isoform compositions in the induced pathologies. Open arrowheads indicate 3R Tau immunobands, and solid arrowheads indicate 4R Tau bands. Asterisks indicate the non-specific immunoreactive bands near 50 KDa, respectively, shown in RD3 and 4RTau blots. Equal proportion of sarkosyl-soluble fractions were loaded, and **f** 15-fold fraction from strain_P1 or **g** 40-fold fraction from strain_P2 mice were loaded as sarkosyl-insoluble fraction. **h** Quantification of **b** from strain_P0; **i** quantification of **d** from strain_P1, and **j** quantification of **e** from strain_P2 induced neuronal tau pathologies, respectively, indicated by AT8, RD3, and RD4. Only ipsilateral hippocampus (iHpx) were quantified for each mouse. Data are presented as mean ± s.e.m. $n = 3$–6 mice per each group. Multiple $t$-tests were performed. Source data is available as a Source Data file.

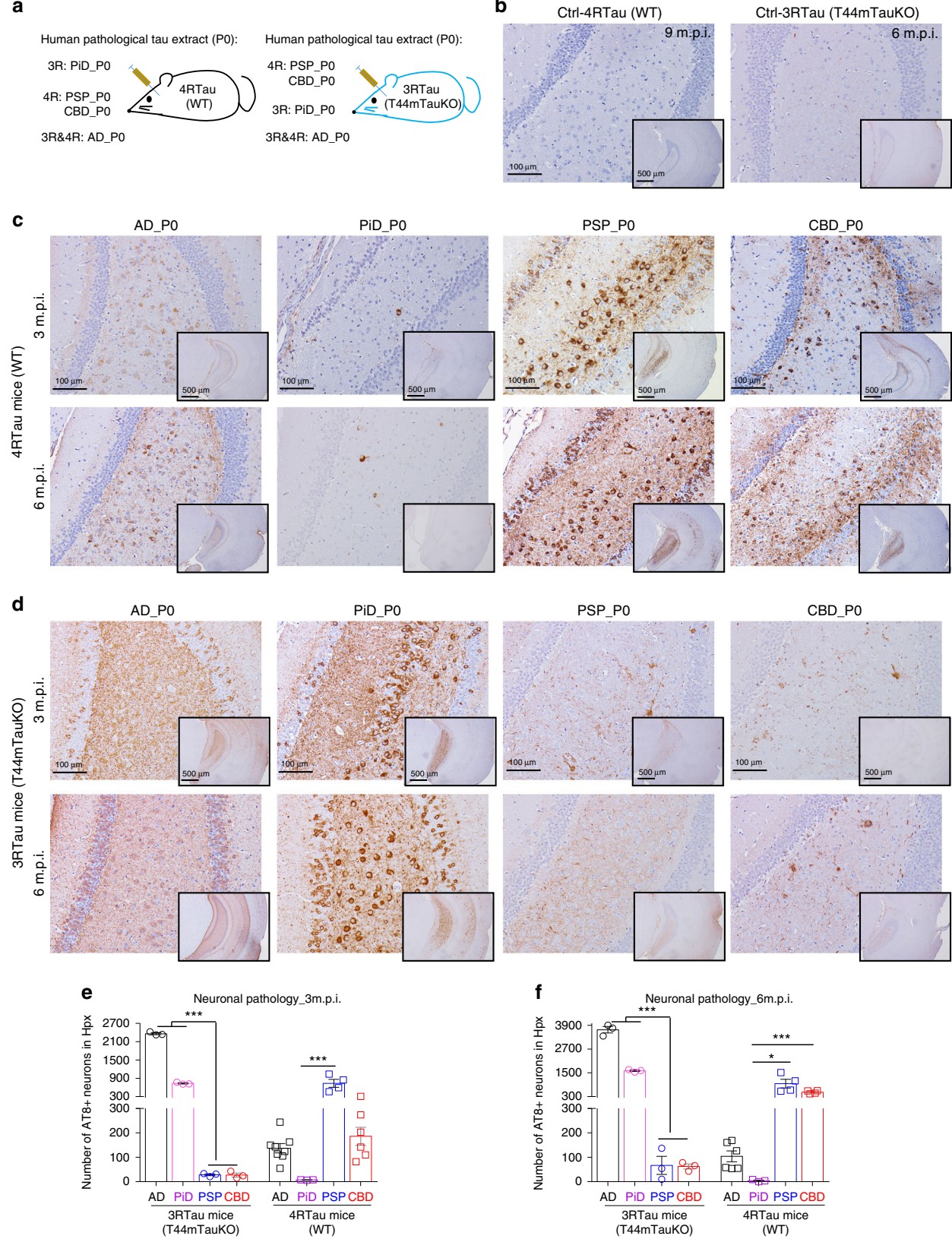

that 4R tau strains are very inefficient in recruiting 3R tau to form pathological tau aggregates. Taken together, these data lead us to ask whether the recruitment of tau isoforms is dependent on the isoform-composition of the strain or if this is an intrinsic strain property independent of isoform-composition.

**Seeding property is independent of seed isoform-composition.** To test whether tau strain isoform compositions determine their, respectively, isoform-specific transmission, we converted the mixed 3R and 4R human brain-derived tau strain AD_P0-tau, into a relatively pure 3R tau strain (AD-3R-tau) or 4R tau strain

**Fig. 3 Tau strain was inefficient in cross-seeding non-corresponding isoform. a** Schematic showing the cross-seeding experiment paradigm whereby different tau strains extracted from human tauopathy brains were injected into the hippocampus and overlying cortex of either 4R tau-expressing (WT) or 3R tau-expressing (T44mTauKO) mice, respectively, to examine whether tau strains could cross-seed non-corresponding tau isoforms. The injection dose for strain_P0 were 1 μg per site. **b** representative IHC staining with MAb AT8 on brain sections from WT mice injected with control brain lysate at 9 m.p.i., and the brain sections from T44mTauKO mice injected with control brain extracts at 6 m.p.i. No significant tangle-like tau pathologies were observed in any of mice injected with the control lysates. **c** Representative IHC staining with AT8 on brain sections from WT mice injected with similar amounts of distinct human tau strains (P0) at 3 and 6 m.p.i. **d** Representative IHC staining with AT8 on brain sections from T44mTauKO mice injected with similar amounts of different human tau strains (P0) at 3 and 6 m.p.i. Note that AD-tau induced most abundant tau pathologies, including massive neuropil-like tau pathologies, which masked the NFTs in the hippocampal DG regions. Inserts in **b**–**d** are images with lower magnifications. Quantification of the AT8-positive cells in WT or T44mTauKO mice, respectively, injected with different human tau strains are shown in **e** for 3 m.p.i. and **f** for 6 m.p.i. time points. Both ipsilateral and contralateral sides were quantified together for each mouse. Data are presented as mean ± s.e.m. $n = 3$–7 in each group, and each dot represents a mouse. One-way ANOVA with Sidak's multiple comparisons tests were performed. $*p < 0.05$; $**p < 0.001$; $***p < 0.001$. Source data is available as a Source Data file.

(AD-4R-tau) through in vivo propagation in mice expressing exclusively 3R (T44mTauKO mice) or 4R (WT mice) tau only, and examined if they behaved differently from human brain-derived AD-tau with all six tau isoforms. In a similar way, we also converted the human brain-derived PiD-tau and PSP-tau into relatively pure tau isoforms comprising PiD-3R-tau and PSP-4R-tau and used them, respectively, as the control for AD-3R- or AD-4R-tau recruitment experiments (Fig. 4a). Since the injected human brain-derived tau proteins were undetectable at 7 days post injection in vivo (Supplementary Fig. 5), the extracted tau pathologies from the abovementioned mice should only be comprised of pathological endogenous mouse-expressing 3R- or 4R-tau isoforms at 3 m.p.i. We also expected that they would maintain the original human strain properties based on the repeated in vivo propagation data shown in Fig. 2. Western blotting confirmed the extracted AD-3R-tau and PiD-3R-tau pathologies in the sarkosyl-insoluble fraction were exclusively 3R tau while the AD-4R-tau and PSP-4R-tau pathologies were exclusively 4R tau (Fig. 4b). If the recruitment of tau isoforms is dependent on the isoform-composition of the seeds, then AD-3R would only seed 3R tau pathologies, while AD-4R would only seed 4R tau pathologies in 6hTau mice, and they would be different from AD_P0. Surprisingly, 3 m.p.i. of AD-3R-tau and AD-4R-tau into 6hTau mice showed both 3R and 4R tau isoforms in the induced pathologies, similar to AD_P0-tau itself. In contrast, PiD-3R-tau predominantly seeded 3R tau and PSP-4R-tau only seeded 4R tau in the 6hTau mice (Fig. 4c, d, f). Biochemical extraction further confirmed AD-3R-tau and AD-4R-tau could seed both 3R and 4R tau, while PiD-3R-tau and PSP-4R-tau could seed only 3R or 4R tau, respectively (Fig. 4e). To further confirm that isoform-composition of the tauopathy seeds does not determine the strain isoform-seeding properties, we injected a similar amount of AD-3R- and PiD-3R-tau into 4R Tau mice. Despite of the fact that both AD-3R and PiD-3R comprises the same 0N3R tau isoform, only AD-3R-tau but not PiD-3R-tau could recruit 4R tau to form tau pathologies in 4R Tau mice. Similarly, AD-4R-tau but not PSP-4R-tau could recruit 3R tau monomers in 3R Tau mice (Fig. 4g–i). Altogether, these data suggest that distinct tau strain isoform-specific seeding is independent of their respective isoform compositions.

**Seeding models recapitulate cell-type-specific tau pathology.** Mapping the regional distribution and cell types of the induced tau pathologies in 6hTau mice by different human tau strains showed AD-tau-induced tau pathologies spread much wider than other tau strains (Fig. 5), but all the tau pathologies were localized in neurons. However, CBD-tau and PSP-tau induced neuronal tau pathology, as well as oligodendrocyte and astrocyte tau pathologies. PiD-tau induced pathologies were predominantly

located in neurons, occasionally a few hyperphosphorylated tau-positive oligodendrocytes were observed (Figs. 5 and 6a–c). Examining the induced tau pathologies in 6hTau mice from 1 to 6 m.p.i. further confirmed AD_P0-tau strain only induced neuronal tau pathologies, whereas PiD_P0-tau also predominantly induced neuronal tau pathologies, although less efficiently compared with AD_P0-tau. PSP_P0-tau and CBD_P0-tau also induced abundant glial tau pathologies in addition to neuronal tau pathologies. Specifically, PSP_P0-tau mainly induced oligodendrocyte tau pathologies, while CBD_P0-tau also induced tau-positive astrocytic plaques, a key pathological feature for CBD (Fig. 6e–g).

In addition to neurons (Figs. 2b, d, e, h–j and 4d, f), the tau isoform compositions of the induced pathologies in glial cells by distinct tau strains were also maintained during in vivo propagation (Fig. 7a–f) and isoform switch experiments (Fig. 7g, h). Altogether, these results suggest the distinct tauopathy strains maintain their cell-type specificity during pathological transmission.

**Gliosis in mice injected with tauopathy strains.** Accumulating data have suggested glial cells are involved in AD and FTLD pathogenesis[19,20]. To examine if gliosis is correlated with the distinct cell-type tau pathologies induced by different tau strains, we respectively examined the time course of astrogliosis by GFAP staining (Supplementary Fig. 6a, b), microgliosis by Iba1 staining (Supplementary Fig. 6c, d) and oligodendrogliosis by measuring white matter regional nucleus (Supplementary Fig. 6e) in the 6hTau mice following injection of 1 μg strain_P0-tau into the hippocampus and overlying cortex. As shown in Supplementary Fig. 6, there were no significant differences in astrogliosis, microgliosis and oligodendrogliosis in CBD/PSP-6hTau mice compared to AD-6hTau or PiD-6hTau mice, suggesting the glial tau pathologies predominantly induced by CBD/PSP-tau may not be due to their specific glial activation.

**Strains maintain their potencies during in vivo propagation.** To further examine whether different tau strains maintained their potencies in seeding pathologies during transmission, we injected 6hTau mice with lysates containing same amounts of pathological P0- or P1-tau (0.2 μg) with equal total amount protein adjusted by adding non-injected 6hTau brain lysates. Following in vivo propagation, distinct tau strains still induce similar amounts of tau pathologies with consistent cell-type specificity (Fig. 8a–d), suggesting the strain potencies are maintained during the in vivo propagation.

To further confirm this observation, we examined the strain potency changes between P0 and P1 in primary neuron cultures, that reliably differentiate tau strains in vitro[13,14]. We observed similar dose response curves for AD and PSP strains and similar

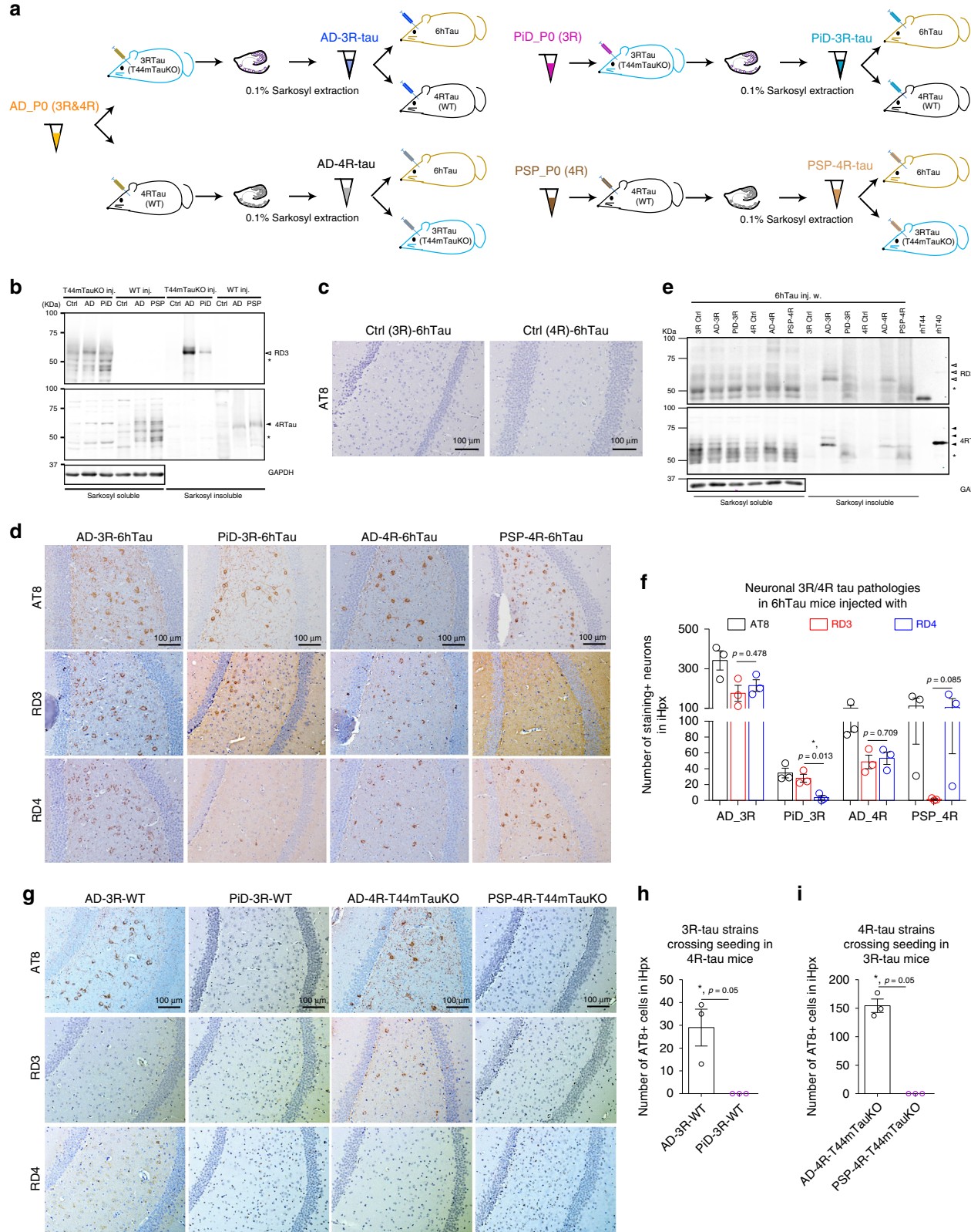

accumulation of insoluble Mtau pathologies by strain_P0 and P1. Moreover, the morphologies of induced tau pathologies by the same strain lineages were comparable (Fig. 8e–g). Owing to the low potency of PiD strain and the very low purity of pathological tau in both CBD_P1 and PiD_P1 (~0.2%), tau aggregates in neurons treated with PiD_P0/P1 cannot be detected. Similarly,

the immune-fluorescence labeled tau aggregates in CBD_P1/P2-treated neurons cannot be quantified due to heavy masking by contaminated proteins.

Altogether, the in vivo and in vitro data consistently support that tau strain properties can be faithfully maintained during in vivo transmission.

**Fig. 4 Isoform-specific seeding pattern is independent of isoform-composition. a** Experimental paradigm showing how human tau strains were converted into AD-3R, AD-4R, PiD-3R, or PSP-4R tau strains through in vivo propagation in mice expressing only 3R tau (T44mTauKO) or only 4R tau (WT). **b** Immunoblots of the transformed tau strains using 3R (RD3) or 4R isoform-selective antibody (4RTau), respectively. Equal proportion of sarkosyl-soluble fractions were loaded, and twofold fraction from T44mTauKO mice or 20-fold from WT mice were loaded as sarkosyl-insoluble fractions. **c** Representative IHC staining with AT8 on brain sections from 6hTau mice injected with 3R tau control lysate extracted from non-injected T44mTauKO mice or 4R tau control lysate extracted from non-injected WT mice at 3 m.p.i. **d** Representative IHC staining with AT8, RD3, and RD4 on adjacent brain sections from 6hTau mice injected with similar amount of AD-3R, AD-4R, PiD-3R, and PSP-4R tau. **e** Immunoblots of the induced tau pathologies in 6hTau mice by different tau strains with isoform-specific tau antibodies. Equal proportion of sarkosyl-soluble fraction and 7.5-fold for AD_3R, 52.5-fold for PiD_3R, 40-fold for AD_4R, and 52.5-fold for PSP_4R induced sarkosyl-insoluble fraction samples were loaded. rhT44 and rhT40 are recombinant human 0N3R and 2N4R tau isoforms, respectively. 3R tau-positive bands, open arrowheads; 4R tau-positive bands, solid arrowheads. Asterisks indicate non-specific blot bands. **f** Quantification of AT8-, RD3- and RD4-positive neurons in the 6hTau mice injected with distinct lysates as shown in **d**. $n = 3$ mice per group. One-way ANOVA with multiple $t$-tests were performed. **g** IHC staining with AT8 in the WT mice injected with similar amounts of AD-3R and PiD-3R tau, and in the T44mTauKO mice injected with similar amounts of AD-4R and PSP-4R tau. Quantification of AT8-positive tau pathologies in **h** AD-3R and PiD-3R tau injected 4R tau mice, and **i** AD-4R and PiD-4R tau injected 3R tau mice. Three mice per group were quantified. Data are presented as mean ± s.e.m. and each dot represents a mouse. One-tailed Mann–Whitney tests were performed. Only ipsilateral hippocampus (iHpx) were quantified for each mouse.

**Neurotoxicity of distinct tau strains**. Since advanced AD and FTLD brains are always associated with tissue atrophy[21,22], we next evaluated the neurotoxicity of distinct tau strains. We examined the effects of strain_P1 on neuron loss, as strain_P1 recapitulated the properties of strain_P0, and were extracted from the same genetic background as 6hTau mice, excluding the cofounding contaminating factors from human brains. As shown in Supplementary Fig. 7a, total NeuN counts in the pathology-enriched hippocampi were similar among the mice injected with distinct P1 tau strains (Supplementary Fig. 7a–c). We further examined the neurotoxicity of different strains in our primary neuron culture system[13,14], by measuring the lactate dehydrogenase (LDH) released from damaged cells and quantify the axon/somatodendrite densities from surviving neurons. As shown in Supplementary Fig. 7d, neurons treated with different doses of AD_P1 did not elicit increased LDH release compared with neurons treated with control lysate. Furthermore, when equal amount of different strain_P1 were added to neurons, LDH release among each group were not significantly different (Supplementary Fig. 7e). Examining the surviving neurons by respectively quantifying nucleus (DAPI), axons (NFL), and somatodendrites (MAP2) staining also did not show significant neurotoxicity of distinct strain_P1 examined (Supplementary Fig. 7g–i). However, since the purity of the strain_P1-tau recovered from injected mouse brains were <1%, a note of caution here is that contaminated proteins could interfere with the interpretation of these results.

**Induced tau pathology reflects distinct strain conformation**. To identify the intrinsic characteristics that distinguish each tau strains, we examined the potential conformational differences among these strains. First, we examined the amount of β-sheet structures induced by the tau strains using Thioflavin S (ThioS) dye. Tau pathologies induced by different strain_P0 in 6hTau mice was similar to their human counterparts and showed consistent different affinities for ThioS (Fig. 9a, b), suggesting the variation of β-sheet structures among different tau strains are propagated in 6hTau mice.

Previous studies demonstrated that a conformational-dependent antibody GT38[23] selectively detects AD-tau pathology among the different human tauopathies (Fig. 9c). When GT38 was used to detect the conformational differences between distinct tau strain-induced tau pathologies, it consistently showed a strong preference in detecting pathologies induced by AD_P0, but not by PSP_P0-tau, CBD_P0-tau and PiD_P0-tau in 6hTau mice (Fig. 9d).

In addition, we previously showed distinct Proteinase K digestion patterns for AD-, PSP-, and CBD-tau[14]. Altogether,

our data further confirmed that substantial conformational differences exist among distinct tau strains, and that these structural differences might be the intrinsic properties of these tau proteins. These structural differences distinguish tau strains from each other in the pathogenic process, leading to distinct transmission patterns with isoform- and cell-type-specificities.

**Discussion**

In this study, we generated a new tauopathy mouse model by inoculating distinct tauopathy brain-derived pathological tau strains into a new mouse line 6hTau that express equal ratio of 3R and 4R tau isoforms, to study the intrinsic properties of Htau strains that are responsible for distinct tau isoform recruitment, cell-type specificity and transmission in human neurodegenerative tauopathies. While the adult WT mice only express 4R tau isoforms, the existing transgenic mice Htau[24] and *MAPT* KI[25] express much more 3R tau than 4R tau. Through virus injection RNA splicing could be modulated in Htau mice, but only in local brain regions that received injection[26]. The transgenic 6hTau mouse generated here is a unique mouse that express all six Htau isoforms with a nearly equal 3R and 4R ratio in many different brain regions during most of mouse lifespan. The equal 3R/4R tau isoform ratio is similar to the tau expression pattern in human adult brains, and enables the 6hTau mouse line to acquire the features of a more authentic human-like model of Htau isoform expression to study these human tauopathies. Through inoculating human disease brain-derived tau strain seeds into brains of the 6hTau mice, we were able to recapitulate distinct disease-relevant tau pathologies with corresponding isoform compositions, as well as cell-type specificities in these mice. Thus, our study provides new insights on the pathogenesis of distinct tauopathies, and implicates conformations but not the isoform compositions as the key intrinsic properties of different tau strains.

How distinct tau strain transmission takes place in different tauopathy is complex. Our findings presented here suggest that the tau transmission process could be divided into three parts: the pathological seeds, the recipient cells and other external/internal risk factors. To dissect such complex issues, we began by focusing on the characterization of the pathological seeds, and we examined how isoform compositions in the seeds affect their transmission pattern. To accomplish our goals, we inoculated distinct strains into the same brain regions, i.e., the hippocampus so as to eliminate confounding factors due to regional and cell-type variations. Our conclusion that strain conformation but not isoform-composition determines the strain properties, is different from previous studies using an in vitro system[27], or engineered cell lines[28], where the authors proposed the need for isoform pairing

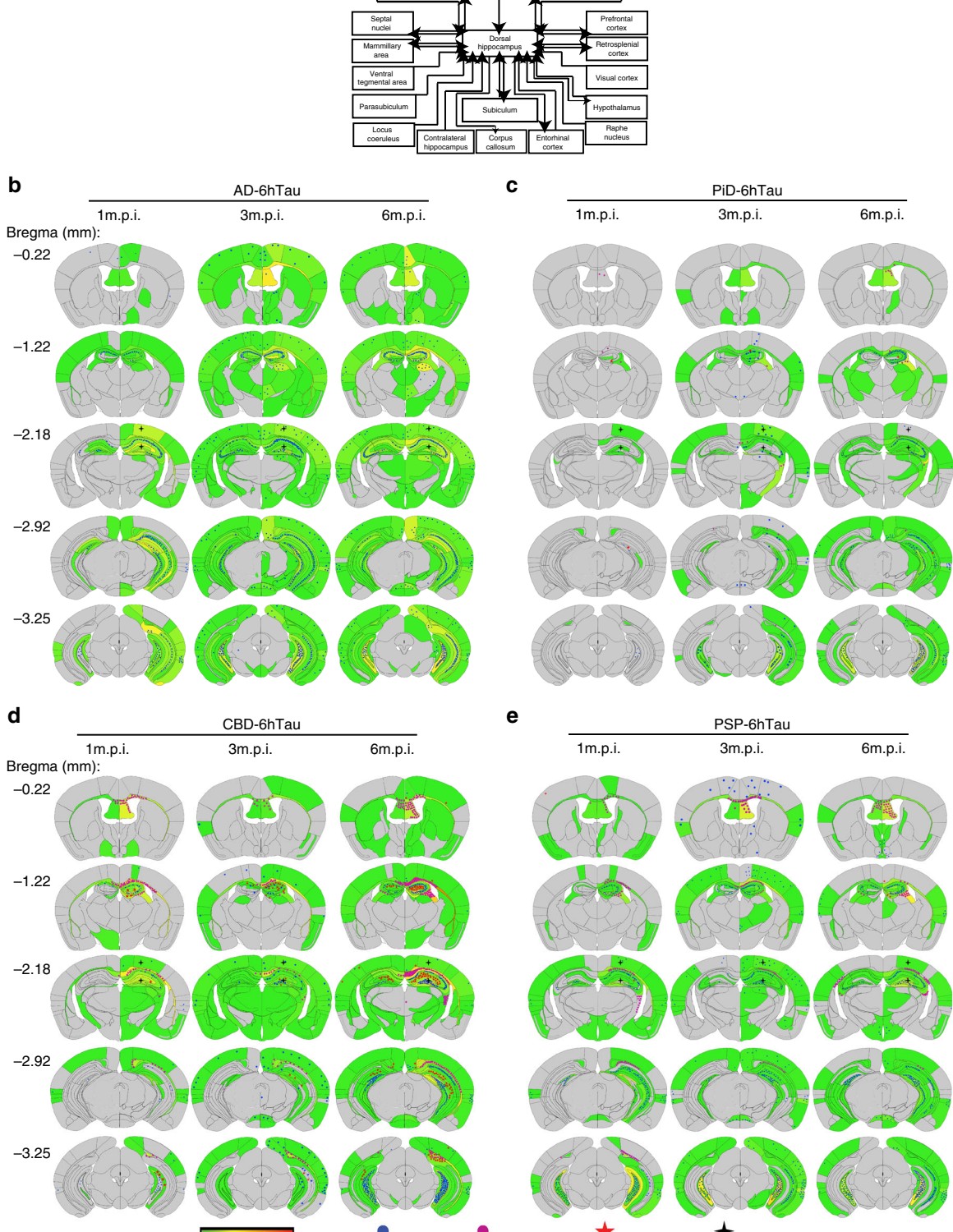

**Fig. 5 Distribution of induced tau pathologies in 6hTau mice. a** Neuroanatomic connectome map showing the anterograde and retrograde connections between the injection site (dorsal hippocampus) and other major brain regions. The distribution of the induced tau aggregates in 6hTau mice by different **b** AD; **c** PiD; **d** CBD, and **e** PSP tau strains were mapped and shown in five representative coronal brain sections at Bregma levels mm: −0.22, −1.22, −2.18, −2.92, −3.25. Neuropil threads were indicated using the heatmap, with gray color indicating no and red color indicating abundant neuropil threads. Neuronal tau pathologies are marked by blue dots, astrocytic tau pathologies are shown as red stars and oligodendrocyte tau pathologies are marked by purple dots. The density of the dots reflects the relative abundance of these tau pathologies. Injection sites are marked with black stars. Mice were examined at 1, 3, and 6 m.p.i., and each group contained 3–6 mice.

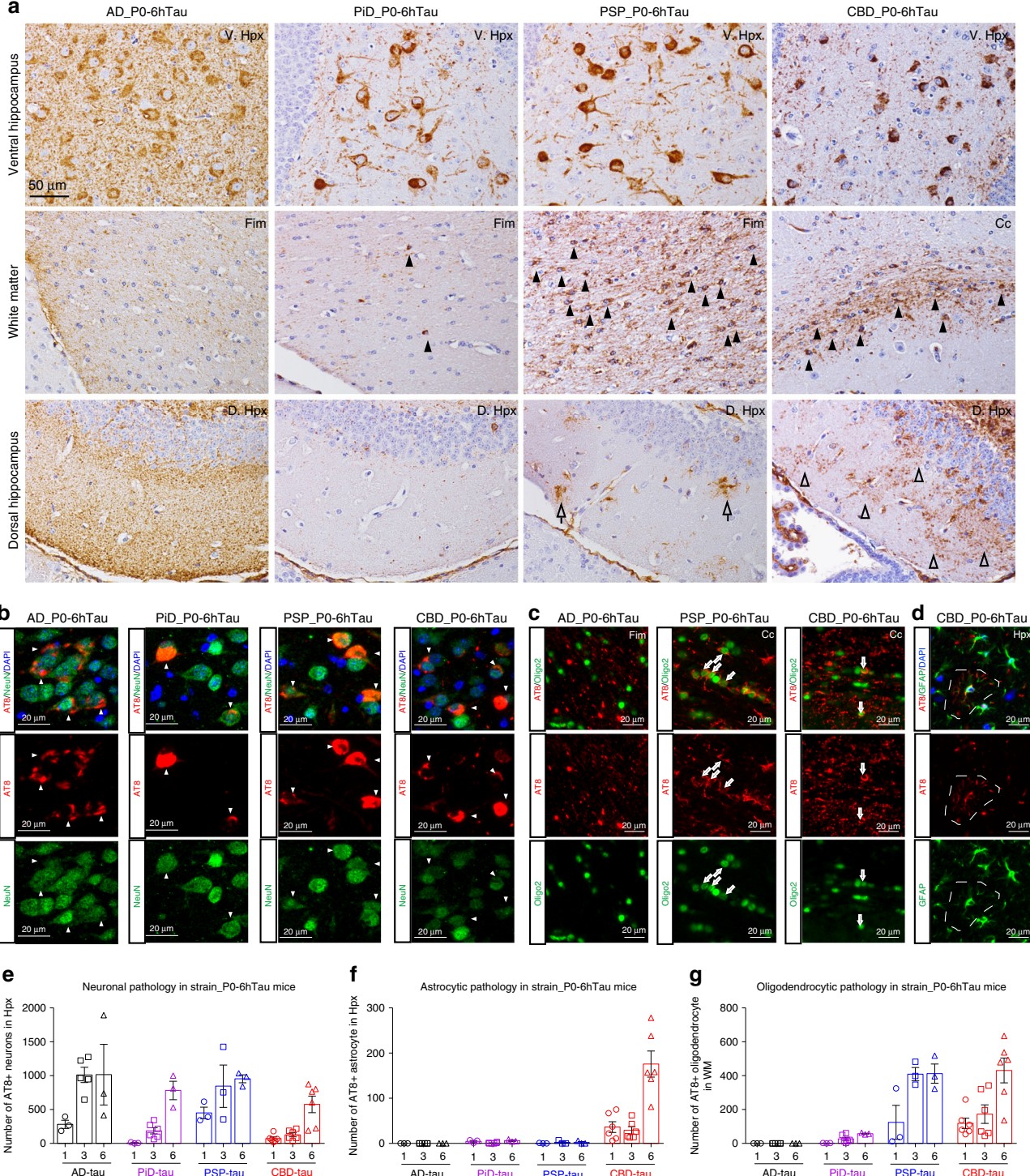

**Fig. 6 Distinct cell-type-specific tau strains were recapitulated in new model. a** Representative IHC staining with AT8 on brain sections from 6hTau mice unilaterally inoculated with distinct human tau strains (P0) at 3 m.p.i. The upper panels showing neuronal tau pathologies in the hippocampus. Middle panels showing white matter with oligodendrocyte tau pathologies induced by PSP_P0 and CBD_P0. The lower panels show tufted-astrocyte-like tau pathologies induced by PSP_P0 and astrocytic plaque-like tau pathologies induced by CBD_P0 in the hippocampus. Oligodendrocytic and astrocytic tau pathologies are indicated by solid and open arrowheads, respectively. Representative double immunofluorescence staining with anti-hyperphosphorylated tau MAb AT8 and **b** neuronal marker NeuN; **c** oligodendrocytic marker, Oligo2 and **d** astrocytic marker GFAP on the pathologies induced, respectively, by strain_P0 in 6hTau mice. Neuronal, oligodendrocytic and astrocytic tau pathologies are indicated by white arrowheads, white arrows and dashed circle, respectively. Quantification of **e** neuronal; **f** astrocytic and **g** oligodendrocytic tau pathologies in the 6hTau mice inoculated with distinct P0 human tau strains at 1, 3, and 6 m.p.i. Both ipsilateral and contralateral sides were quantified together for each mouse. Data are presented as mean ± s.e.m.. $n = 3$–6 for each group, and each dot represents a mouse subjected to quantitation. WM white matter, Fim fimbria, Cc corpus callosum, Hpx, hippocampus. Source data is available as a Source Data file.

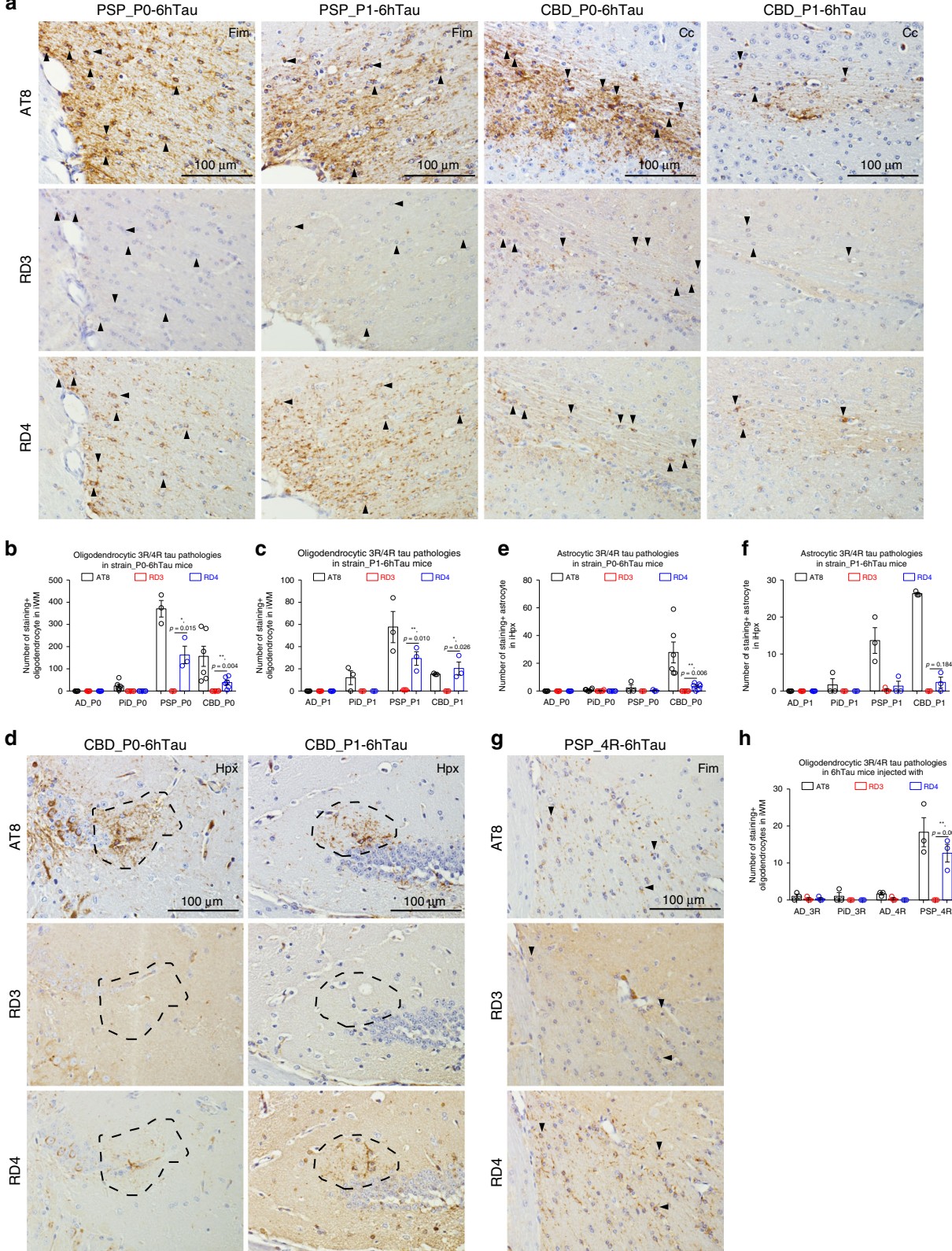

between the seeds and the recipient monomers. This could reflect the fact that recombinant K18/K19 seeds used in these previous studies could not fully recapitulate disease-associated strain conformations, which have been recently demonstrated for AD and PiD filaments by cryo-EM[29,30]. Furthermore, seeding of tau fibrils in cell lines expressing truncated tau fragments[28] is not

equivalent to conducting these experiments in neurons expressing all six Htau isoforms under the human *MAPT* gene.

To gain a better understanding of the underlying mechanism of tau propagation, it is also important to examine the recipient cells to determine whether the tau expression pattern in different brain regions and cell types would correlate with the distinct

**Fig. 7 Tau isoform compositions in the induced glial pathologies.** Representative IHC staining of induced (**a**) oligodendrocytic and **d** astrocytic tau pathologies induced in 6hTau mice injected with P0- and P1-tau as shown in Fig. 2b and d using AT8, RD3, and RD4 antibodies on adjacent brain sections. Quantification of induced oligodendrocytic tau pathologies in **b** strain_P0- and **c** strain_P1–6hTau mice. Quantification of induced astrocytic tau pathologies in **e** strain_P0- and **f** strain_P1-6hTau mice. n = 3–6 mice per group were quantified, and multiple t-test were performed. **g** representative IHC staining and **h** quantification of oligodendrocytic tau pathologies in PSP_4R-6hTau mice as shown in Fig. 4d. n = 3 mice per group were quantified, and multiple t-test were performed. Quantifications were from ipsilateral brain side (**i**) of each mouse. Data are presented as mean ± s.e.m. Significant differences were considered when p ≤ 0.05. WM white matter, Fim fimbria, Hpx hippocampus. Solid arrowheads indicate oligodendrocytic pathologies, and dashed circles indicate astrocytic pathologies. Source data is available as a Source Data file.

pathological tau transmission. Although it is beyond the scope of the current study, we also examined tau expression in terms of total tau levels and isoform compositions in normal human brains and 6hTau mice. We found that the overall tau expression patterns are comparable among different brain regions, except that in addition to the six human tau isoforms, the big tau isoform was also detected in spinal cord in both human and 6hTau mice. Since tau inclusions are usually not present in the spinal cords of AD or FTLD patients[31], it is interesting to determine in future studies if phosphorylated dependent and independent big tau isoforms affect tau transmission. Furthermore, it will be desirable to elucidate the exact tau isoform expression pattern in different cell types in the brain, since such valuable information will facilitate our understanding of the pathological cell-type-specificity in distinct tauopathies.

It is notable that the pathologies induced in PiD_P0-T44mTauKO was higher than that in PiD_P0-6hTau mice. At least two factors need to be considered. First, PiD_P0 is a 3R tau strain, and recruits 3R monomers. As shown in Fig. 1b, d, the total tau expression level in 6hTau mice is roughly 1.5-fold higher than T44mTauKO mice. This means in 6hTau mice, the 3R tau isoform expression was only 75% of that in T44mTauKO mice. The reduced 3R monomer tau expression might explain why lower amounts of tau pathologies were observed in 6hTau mice compared with T44mTauKO mice induced by the same dose of PiD_P0. Second, unlike T44mTauKO mice, 4R tau are also expressed in 6hTau mice. Future studies will elucidate if 4R tau interfere/modulate 3R tau fibril assembly similar to the role played by 3R on 4R tau fibril assembly[32,33].

Pathological cell-type-specificity is another key feature of distinct tauopathy strains, which has not been recapitulated in any existing in vitro or cell-based systems. We recently succeeded in recapitulating the distinct cell-type specificities in WT mice[14], however, due to the different messenger RNA splicing mechanisms between human and mouse, WT mice could only be used to study 4R-tau containing strains, like AD, PSP, CBD, but not 3R-tau strain PiD. Using our new 6hTau mice, we could study any kind of human tau strains in the new mouse models, in which the cell-type specificities of the different human tau strains could not only be recapitulated but also propagated with fidelity over serial passaging in vivo (Figs. 2, 6, and 8). Thus, the 6hTau mice developed in our current study generated data that are more relevant to human tauopathies.

Based on our findings, we hypothesize that initial tau pathogenesis could be divided into two phases: the initial misfolded seed formation phase followed by amyloidogenic amplification phase, where the endogenous monomeric tau isoforms are templated by misfolded seeds to adopt misfolding conformations followed by transformation into pathological forms. Multiple studies[14,34] including this current work have demonstrated that human tau strain seeds carry distinct conformations, but it is unclear how distinct tau strain seeds are initially formed. A recent study on different alpha-synuclein strains suggests unique cellular milieu plays a critical role in distinct strain pathogenesis[35]. The conformational differences among the initial tau strain seeds

would then lead to differential accessibility for different isoform monomers, resulting in different isoform recruitment. This possible mechanism is supported by recent Cryo-EM data[29,30]. If non-pairing isoform monomers are recruited, it would cause instability in the assembly reaction, leading to the fibril structure breakdown (Supplementary Fig. 8). Despite this knowledge, it is unclear how different cell types provide distinct environments to initiate strain seeding conformation and if cell-type-specific co-factors are involved in such processes.

Tau strains that undergo transmission have distinct cell-type specificities, and we demonstrated this property is maintained during serial in vivo propagation, suggesting the cell-type transmission specificity is an intrinsic feature of the tau strains. Since CBD-tau or PSP-tau largely exist in glial cells and they preserve their specificity for oligodendrocytes and astrocytes through transmission in mouse models, this suggests that certain modulators may exist either on the surface or inside the glial cells that actively mediate cell-type-specific transmission of the corresponding tau strains. Identifying such modulators will be very critical to elucidate distinct tau strain transmission mechanisms and to develop treatments for these devastating diseases.

## Methods

**Animals**. In this study, 215 6hTau mice (C57B6 genetic background, from four generations), 75 T44mTauKO mice (C57BL/6 genetic background, from three generations), 85 WT mice (C57BL/6 genetic background), 6 T44 mice (B6D2F1 genetic background, from one generation), and 6 hT-PAC mice (C57B6 genetic background, from one generation) were used. 6hTau mice were generated by crossing a Tg mouse line expressing the normal Htau gene *MAPT* on a Mtau gene *Mapt* KO background (hT-PAC-N+/+;*Mapt*−/−)[7] with a Tg mouse line expressing the human tau gene *MAPT* bearing a mutation at exon 10 + 14 on Mtau gene *Mapt* KO background (hT-PAC-E10+14+/−;*Mapt*−/−). This mouse line was maintained as hT-PAC-N+/+;E10+14+/−;*Mapt*−/−. 6hTau genotyping was performed by PCR using primers: Htau-E10-S1: 5′-ccaagtgtggctcaaaggat-3′, Htau-E10-A1: 5′-cgcactcacaccacttccta-3′. The PCR products were then digested with BmgBI to distinguish E10+ 14 mutation. Internal-positive control primers: Int_pos_cont_Htau_F: 5′-ctaggccacagaattgaaagatct-3′, Int_pos_cont_Htau_R: 5′-gtaggtggaaattctagcatcatcc-3′.

T44mTKO Tg mice were generated by crossing mouse line T44[18] with Mtau KO mice[36], resulting in the new line expressing only 0N3R Htau. The line was maintained as T44+/−;*Mapt*−/−, and the genotyping primers are: MoPrp S: 5′-tggctcgggacttcaaaatcag-3′, MoPrp AS: 5′-tccccagcctagacacgagaat-3′. For tau knockout genotyping, TauKO E1_F: 5′-gccagaggccacttgtgtag-3′; WT_F: 5′-aatggaagaccatgctggag-3′; Shared TauKO/WT_R: 5′-attcaacccctcgaatttt-3′.

Non-Tg C57BL/l mice were either purchased from Charles River or obtained from a breeding colony maintained in our center.

Both male and female mice were used in these studies. All animals were allocated randomly for histological experiments, but groups were counterbalanced for animal sex and group average body weight. To avoid the potential genetic drifts, we always injected mice from the same generation with the different tau strains to compare their strain properties. All animal protocols were approved by the University of Pennsylvania's Institutional Animal Care and Use Committee (IACUC). We have complied with all relevant ethical regulations for animal testing and research.

**Purification of pathological tau from postmortem human brain**. Human brain tissues from two cases of each AD, PiD, CBD and PSP with abundant frontal cortical tau pathologies and one normal case were included in this study (See Supplementary Table 1). All cases except one PSP were from CNDR brain bank and diagnosed based on accepted neuropathology criteria[37,38]. Frozen frontal cortex tissues were used to extract different human tauopathy pathologies, and the

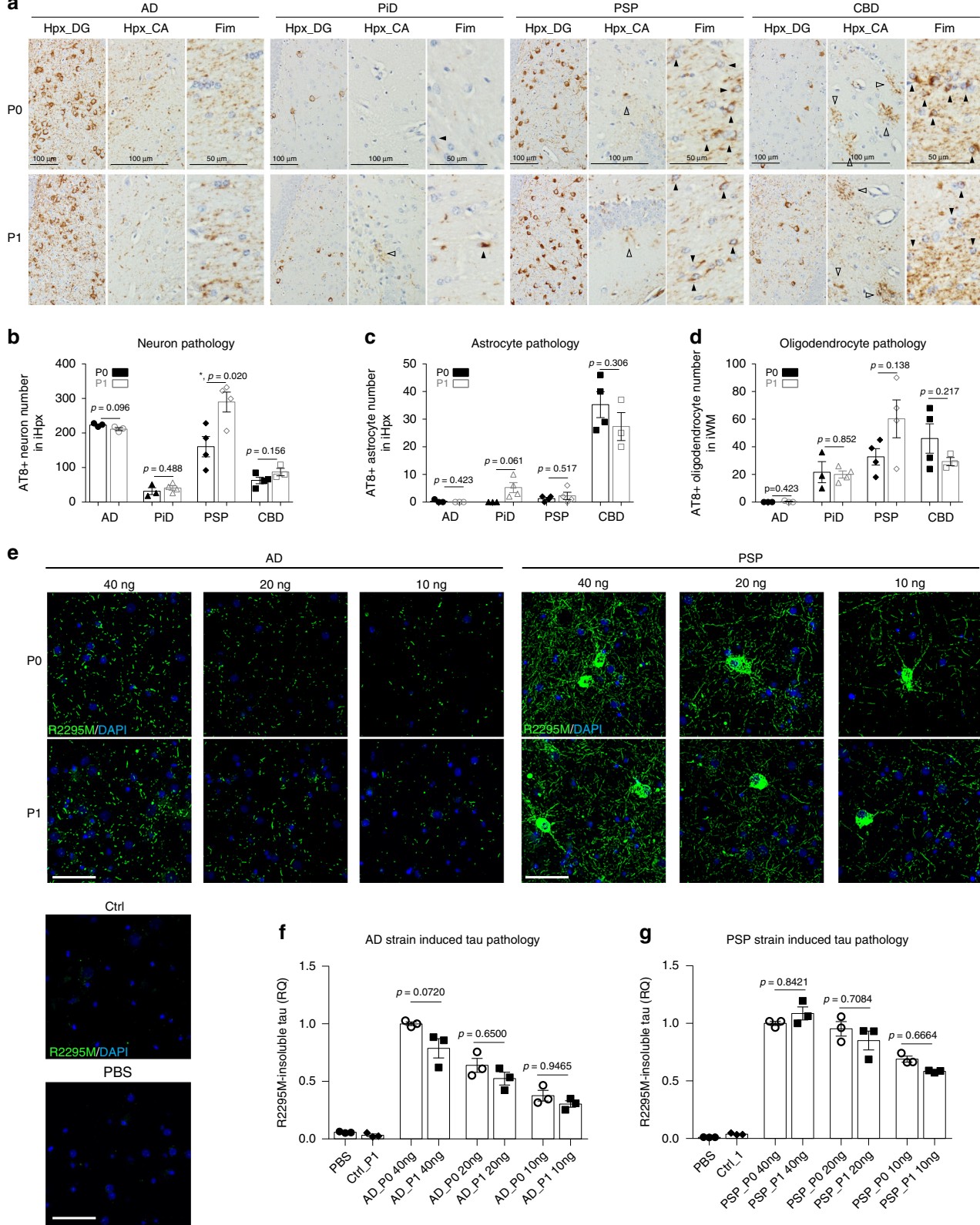

purification method was similar to AD-tau isolation method we reported recently[13,14,16], except for PSP and CBD, where we used both gray and white matter for extractions. Briefly, selected tauopathy brain tissues were homogenized in nine volumes (v/w) of high-salt buffer [10 mm Tris with 0.8 m NaCl, pH 7.4] with 0.1% Sarkosyl and 10% sucrose added, and centrifuged at 10,000 × g for 10 min at 4 °C. Pellets were re-extracted twice using the same high-salt buffer and the supernatants from all three extractions were filtered and pooled. Additional Sarkosyl was added to the pooled supernatants to reach 1% and the samples were

rotated for 1 h at room temperature. The samples were centrifuged at 300,000 × g for 60 min at 4 °C and the resulted 1% Sarkosyl-insoluble pellets containing pathological tau were resuspended in phosphate-buffered saline (PBS). The resuspended Sarkosyl-insoluble pellets were further purified by a brief sonication using a handheld probe (Qsonica XL-2000), followed by centrifugation at 100,000 × g for 30 min at 4 °C. The pellets were resuspended in PBS at 1/2 to 1/5 of the pre-centrifugation volume, sonicated, and spun at 10,000 × g for 30 min at 4 °C to remove large debris. The final purified supernatants contained insoluble,

**Fig. 8 Tau strains maintained their potencies during in vivo propagation. a** Representative IHC staining with AT8 on brain sections from 6hTau mice unilaterally inoculated with 0.2 μg tau strain extracted either from primary human diseased brains (P0) or strain_P0 injected 6hTau mice (P1). The total protein in P0 and P1 lysates were also adjusted the same using non-injected 6hTau brain lysates. The DG region of ventral hippocampus showed neuronal tau pathologies in all the cases. The astrocytic and oligodendrocytic tau pathologies induced by PSP and CBD strains are, respectively, shown in the hippocampal CA regions and the fimbria white matter region. Quantification in **b** neuronal; **c** astrocytic, and **d** oligodendrocytic tau pathologies in the ipsilateral sides (**i**) of 6hTau mice injected with strain_P0/P1 shown in **a**. Two-tailed $t$-tests were performed with significant difference where $p \leq 0.05$. Data are presented as mean ± s.e.m. and each dot represents a mouse. Hpx hippocampus, DG dentate gyrus, CA cornu ammonis, Fim fimbria, WM white matter. Quantifications were from ipsilateral (**i**) sides. **e** Representative ICC staining with anti-mouse tau antibody R2295M on primary mouse neurons treated with AD_P0/P1 or PSP_P0/P1 at three dosages (10, 20, and 40 ng) per well. Control (Ctrl) brain lysate was extracted in the same way as strain_P1 from non-injected old 6hTau mouse brains. The amount of Ctrl lysate was matched to the total protein of AD_P1 40 ng, which contains the highest total protein. AD_P0/P1 all induced tau aggregate in the neuropil, while PSP_P0/P1 also induced obvious tau aggregate in somas. Scale bar, 50 μm. Relative quantification in **f** AD strains and **g** PSP strains. The normalization was made to DAPI counts and then, respectively, to AD_P0 40 ng dose for AD strains, or PSP_P0 40 ng dose for PSP strains. Data are presented as mean ± s.e.m. and each dot represents a batch repeat. One-way ANOVA with multiple comparisons were performed. Significant differences were considered when $p \leq 0.05$. Source data is available as a Source Data file.

pathological tau, and were identified as AD_P0-tau, PiD_P0-tau, CBD_P0-tau and PSP_P0-tau in subsequent experiments. The total protein concentrations of the final supernatant fractions were analyzed by bicinchoninic acid (BCA) assay (Fisher, Cat. 23223/23224), tau concentrations were estimated by western blotting with recombinant tau protein as the standard. The concentration of AD-tau in each preparation was 1.6–2.3 μg/μl, with a purity of 21.6–29.2%, while concentrations of PSP-tau, CBD-tau and PiD-tau were 0.1–0.56 μg/μl, with a purity of <1%. The concentration of tau from control brains in each preparation was 2–5 ng/ml, with a purity of 0.016–0.049%. The concentration of Aβ42 detected by enzyme-linked immunosorbent assay (ELISA) ranged from undetectable to 0.01 μg/ml, Aβ40 was 0.002–0.004 μg/ml and α-syn was 5.8–7.6 μg/ml.

**Extraction of pathological tau from mouse brain**. Mouse brains were removed from mice at designated time points and different regions, including hippocampus and cortex were dissected separately, snap-frozen in dry ice, and then stored at −80 °C before extraction. To extract the induced pathological tau from mouse brains, hippocampal tissues from injected mice were homogenized with 9 volumes of PBS containing 0.1% sarkosyl, proteinase inhibitor cocktail and phenylmethylsulfonyl fluoride (PMSF), the homogenates were then centrifuged at 10,000 × g for 10 min, the low-spin pellets were discarded, and the supernatants were further centrifuged at 100,000 × g for 45 min. The higher spin pellets were then washed twice with PBS and finally resuspended in a small volume of PBS for injection. All PBS used in our experiments was $Ca^{2+}$- and $Mg^{2+}$- free. The concentration of insoluble tau extracted from different mouse brains were 2–294 ng/μl, with a purity of 0.01–1.85%.

**Recombinant tau purification and in vitro fibrillization**. Recombinant human tau isoforms used in these studies were expressed in BL21 (DE3) RIL cells (Agilent Technologies, Cat. 230245) and purified by cationic exchange using a Fast Protein Liquid Chromatography (FPLC) as previously described[39]. Briefly, bacteria were grown in TB media containing antibiotics at 37 °C and induced with isopropyl-β-d-thiogalactopyranoside at a final concentration of 0.8 mM when the OD reached 0.6. After agitation for 2 h, cells were harvested by centrifugation. The pellet was resuspended in high-salt RAB buffer [0.1 M MES, 1 mM EGTA, 0.5 mM MgSO₄, 0.75 M NaCl, 0.02 M NaF, 1 mM PMSF, 0.1% protease inhibitor cocktail (100 μg/ml each of pepstatin A, leupeptin, TPCK, TLCK, and soybean trypsin inhibitor and 100 mM EDTA), pH 7.0] and homogenized. The cell lysates were heated to 100 °C for 10 min, rapidly cooled on ice for 20 min and centrifuged at 70,000 × g for 30 min. Supernatants were dialyzed into FPLC buffer A [20 mM piperazine-N,N′-bis (2-ethanesulfonic acid), 10 mM NaCl, 1 mM EGTA, 1 mM MgSO₄, 2 mM DTT, 0.1 mM PMSF, pH 6.5], applied onto a HiTrap Sepharose HP IEX cation-exchange column (GE Healthcare Cat. 17–1154–01), and eluted with a 0−0.4 M NaCl gradient using an ÄKTA FPLC system (GE Healthcare). The fractions were checked for the presence of the tau proteins by sodium dodecyl sulfate−polyacrylamide gel electrophoresis (SDS−PAGE) followed by Coomassie Blue R-250 staining. Those containing the desired tau profile were pooled together and dialyzed against 100 mM sodium acetate buffer, pH 7.0. Proteins were concentrated using an Amicon Ultra-15 Centrifugal Filter Unit (Millipore Corp. Cat. UFC905024).

Tau PFFs were made as previously described[15]. Briefly, 40 μM recombinant Tau were mixed with 40 μM low molecular weight heparin and 2 mM DTT in 100 mM sodium acetate buffer (pH 7.0). The proteins were incubated at 37 °C with constant agitation at 1000 rpm for 5–7 days. Successful fibrillization was confirmed using thioflavin T fluorescence assay, sedimentation tests, and negative stain electron microscopy. Before being used for seeding experiments, fibrillization mixture was centrifuged at 100,000 × g for 30 min at 4 °C. The resulting pellet was resuspended in equal volume of 100 mM sodium acetate buffer (pH 7.0), without heparin and DTT, and frozen as single use aliquots at −80 °C.

**Sandwich ELISA**. The concentrations of tau, α-syn, Aβ40, and Aβ42 in AD-tau preparations were measured using sandwich ELISA as previously described[35,40,41].

Briefly, for quantifying tau, lysates from sequential extraction were diluted 1000 times in EC buffer [0.02 M sodium phosphate buffer (pH 7.0), 2 mM EDTA, 0.4 M NaCl, 0.2% BSA, 0.05% CHAPS, 0.4% Blockace, 0.05% NaN₃]. Serially diluted recombinant human T40 was used to generate standard curves. Thirty microliters of diluted samples were loaded into each well of 384-well Nunc Maxisorp (Fisher, Cat. 12565347) clear plate coated with capture antibody Tau 5. After overnight incubation at 4 °C, captured proteins were reported using biotin-labeled BT2 and HT7, which were left to incubate overnight at 4 °C. The next day, following 1 h of incubation with HRP-conjugated streptavidin at 25 °C, the plate was developed using trimethylbenzidine peroxidase substrate solution.

To measure the concentration of α-Syn, 384-well plates were coated with 100 ng (30 μl per well) Syn9027, a MAb to α-Syn, in Takeda buffer and incubated overnight at 4 °C. The plates were washed 4 × with PBS containing 1% Tween 20 (PBS-T), and blocked using Block Ace solution (90 μl per well) (AbD Serotec, Cat. BUF029) overnight at 4 °C. Then, the plates were incubated with brain lysates at 4 °C overnight using recombinant α-Syn monomer as standards. The plates were then washed with PBS-T and a rabbit monoclonal anti-α-Syn antibody, MJF-R1 (1:1,000, 30 μl per well) was added to each well and incubated at 4 °C overnight. After washing, goat-anti-rabbit-IgG conjugated to horse radish peroxidase (Cell Signaling Technology, 1:15,000, 30 μl per well) was added to the plates followed by incubation for 2 h at room temperature. Following another wash, the plates were developed for 10–15 min using 1-Step Ultra TMB-ELISA substrate solution (30 μl per well, Thermo Fisher Scientific, Cat. 34029), the reaction was quenched using 10% phosphoric acid (30 μl per well) and plates were read at 450 nm on a Molecular Devices Spectramax M5 plate reader.

The combinations of capture and reporting antibodies for other protein are: Ban50/BA27 for Aβ40, Ban50/BC05 for Aβ42.

**Tau protein dephosphorylation**. Mouse cortical tissues were homogenized in 4 volume RAB high-salt buffer [0.75 M NaCl, 100 mM Tris, 1 mM EGTA, 0.5 mM MgSO4, 0.02 M NaF, 2 mM DTT, containing protease inhibitor cocktail, and PMSF but no phosphatase inhibitor, pH 7.4]. The homogenates were centrifuged at 100,000 × g for 40 min at 4 °C. The supernatant was boiled at 100 °C for 5 min, and then centrifuged at 10,000 × g for 20 min at 4 °C. The supernatant was then treated with lambda protein phosphatase (New England Biolabs, Cat. P0753S) at 37 °C for 3 h and probed by tau antibodies using western blot.

**Stereotaxic surgery**. Most of the mice (WT, 6hTau and T44mTauKO) were injected at the ages between 3 and 5 months unless specified. Mice were deeply anesthetized with ketamine/xylazine/acepromazine, immobilized in a stereotaxic frame (David Kopf Instruments), and aseptically inoculated with human brain extracts in the dorsal hippocampus and/or overlying cortex of one hemisphere (bregma: −2.5 mm; lateral: +2 mm; depth: −2.4 mm and −1.4 mm from the skull) as previously described[13,14,16]. For the synthetic tau PFFs, or in vivo mouse propagated materials, they were only injected into the dorsal hippocampus. To examine the spreading pattern of pathologies in vivo, all the mice were unilaterally injected in the right hemisphere, with a total of 2 μg of pathological tau strains. For in vivo propagation experiment, 6hTau mouse were bilaterally injected, with control lysate or 0.1–0.5 μg P1-tau or 0.01–0.03 μg P2-tau per side. For the tau strain cross-propagation injection experiments, the hippocampus in each mouse were bilaterally injected with control lysates or 0.1 μg AD-3R, AD-4R, PiD-3R, or PSP-4R tau.

**Immunostaining**. Mice were sacrificed and processed as previously described[16], while immunostainings were performed on 6 μm paraffin embedded mouse brain sections as described previously[15]. Most of the antibodies worked in 10% formalin-fixed mouse brain tissues except GT38, which had to be performed

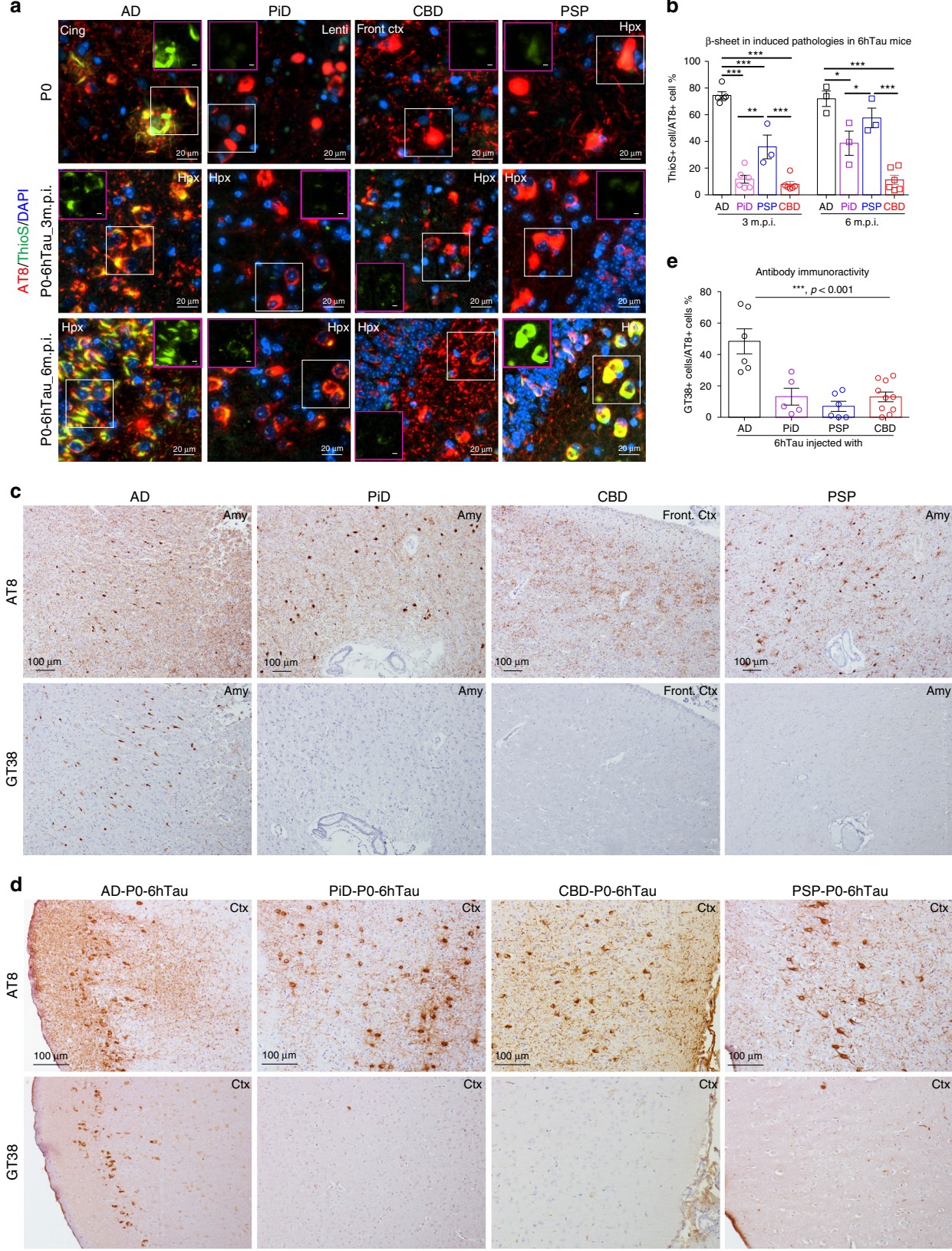

on 70% ethanol-fixed mouse tissues. The quantifications of pathological tau-positive cells were blinded and manually counted from eight coronal sections (Bregma 0.98, −0.22, −1.22, −2.18, −2.92, −3.52, −4.48, and −5.52 mm) for each mouse based on morphology. Human brain slices fixed in formalin or ethanol were also sectioned into 6 μm and stained with antibodies as indicated in the study.

For NFL (neurofilament light chain, 1:1000, Rabbit, homemade) and MAP2 (1:2000, AB5543, Millipore) staining, primary hippocampal neurons were fixed with 4% PFA at room temperature (RT) for 20 min and permeabilized with 0.01% Triton-X for 10 min. For insoluble tau, neurons were first extracted with 1% hexadecyltrimethylammonium bromide (Sigma-Aldrich) to remove soluble tau and fixed with 4% PFA afterward. A mouse tau-specific antibody (R2295M, 1:2000,

**Fig. 9 Distinct tau strains harbored distinct conformations. a** Representative immunostaining with anti-hyperphosphorylated tau MAb AT8, the β-sheet amyloid dye Thioflavin S and nuclear DAPI stain on brain sections from: upper panel, the different human tauopathy brains used for extracting human tau strains; middle and lower panels, the AD_, PiD_, CBD_ and PSP_P0-6hTau mice at 3 and 6 m.p.i., respectively. Purple frame inserts show the Thioflavin S staining in white frame areas. Scale bars in the purple frames represent 5 μm. The differences in the amounts of β-sheet structure indicated by the percentage of Thioflavin S-positive pathologies in 6hTau mice injected with distinct human tau strains at two different time points were quantified in **b**. Data are presented as mean ± s.e.m. $n = 3–6$ for each group, with each dot representing a mouse. One-way ANOVA with Sidak's multiple comparisons tests were performed. *$p < 0.05$; **$p < 0.001$; ***$p < 0.001$. **c** IHC staining on adjacent human tauopathy brain sections using MAbs AT8 and conformational-dependent GT38 specific for AD tau[23]. **d** Representative IHC staining with AT8 and GT38 MAbs in 6hTau mice and **e** quantification of the paired adjacent brain sections from 6hTau mice injected with distinct tau strains. The amount of GT38 MAb AD-like "conformation" was determined by the percentage of GT38-positive pathologies among the 6hTau mice injected with distinct human tau strains. Data are presented as mean ± s.e.m. One-way ANOVA was performed for statistical analysis. Cing cingulate, Lenti lentiform, Front ctx frontal cortex, Hpx hippocampus, Amy amygdala. Source data is available as a Source Data file.

Rabbit, homemade) was used to reveal mouse tau pathology. Primary antibodies were incubated overnight revealed by incubation of Goat anti-Rabbit and Goat anti-Chicken secondary antibodies. DAPI (Sigma-Aldrich) was used for nuclei. After staining cells were scanned by IN Cell Analyzer 2200 (GE Healthcare Life Sciences) and analyzed by HALO software (Indica Labs) for intensity and area of occupancy.

**Gallyas silver staining**. After deparaffinized and hydrated, sections were put in 5% periodic acid for 5 min, and then washed twice in distilled $H_2O$ (d$H_2O$) for 5 min each time. The sections were incubated with silver iodide solution (12 g sodium hydroxide, 30 g potassium iodide and 10.5 ml 1% silver nitrate, add d$H_2O$ to 300 ml) for 1 min and placed in 0.5% acetic acid for 5 min twice. After rinsing in d$H_2O$, the sections were developed in developer solution mixed by solution A, B and C. Solution A (50 g anhydrous sodium carbonate in 1000 ml d$H_2O$); Solution B (1.9 g ammonium nitrate, 2 g nitrate, 10 g tungstosilicic acid in 1000 ml d$H_2O$); Solution C (1.9 g ammonium nitrate, 2 g silver nitrate, 10 g tungstosilicic acid, 7.6 ml 37% formaldehyde in 1000 ml d$H_2O$). The reaction was stopped in 0.5% acetic acid for 5 min.

**Histology heatmaps**. Semi-quantitative analyses were performed on mouse brain sections as previously described[16]. Average scores from each group were imported into customized software to create heat maps representing the abundance and distributions of neuropil thread tau pathologies. On top of these heatmaps, tau strain-induced distinct cell-type-specific pathologies were then respectively marked on the heatmaps.

**Primary neuron culture and brain lysate transduction**. Transduction of tau strains was done as previously described[13]. Briefly, CD1 mouse hippocampal neurons were prepared from embryos 16–18 days. The dissected hippocampal neurons were incubated with papain (Worthington Biochemical Corp., Cat. LS003126) at 37 °C for 45 min. Cells were triturated thoroughly and plated on PDL (poly-D-lysine, Sigma-Aldrich, Cat. P0899) coated 96-well plates (Perkin Elmer LLC VIEWPLATE-96 Black, Fisher Scientific, Cat. 50–905–1605) with a density of $1.75 × 10^4$ cell/100 μl/well. Ten percent FBS was added into the medium at DIV0 and replaced with fresh medium 24 h later (DIV1). Mouse and human brain lysates were diluted with dPBS and sonicated with a bath sonicator for 30 min (30 s pulse and 30 s break/min, Bioruptor®, Diagenode, Cat. B01060010) at high intensity. Neurons were treated with lysates from DIV7 to DIV21 for 14 days. The amount of Tau or total protein is indicated in figure legends. LDH assay (Pierce™ LDH Cytotoxicity Assay Kit, Thermo Fisher Scientific, Cat. 88953) was done following the manufacturer's instruction.

**Western blotting**. For western blotting, equal proportion of homogenates or sarkosyl-soluble fractions from each experimental group and 20–35 times sarkosyl-insoluble fractions were separated using SDS-PAGE gels, and transferred to 0.22 μm nitrocellulose membranes. The nitrocellulose membranes were then blocked in Odyssey blocking buffer (Li-Cor Biosciences, Cat. 927–40000) or 5% milk diluted in tris-buffered saline (TBS) before being immunoblotted with specific primary antibodies (Supplementary Table 2). The blots were further incubated with IRDye-labeled secondary antibodies and scanned using ODY-2816 Imager (Li-Cor Biosciences). The optical densities were measured with Image Studio software (Li-Cor Biosciences). The uncropped and unprocessed scans of all the important blots shown in this paper are displayed in Supplementary Fig. 9.

**Statistics**. Measurements were from distinct samples. Two-tailed *t*-test was performed when two groups were compared, one-way *ANOVA* or multiple *t*-tests were performed when multiple groups were compared, according to different conditions. Statistical analyses were performed using Prism software (GraphPad Software, Inc). Significant differences were considered when *$p ≤ 0.05$; **$p ≤ 0.01$,

and ***$p ≤ 0.001$. Sample size was determined based on previous experience. Detailed statistical information for all the data presented in this study is listed in Supplementary Table 3.

**Reporting summary**. Further information on experimental design and analysis is available in the Nature Research Reporting Summary linked to this article.

## Data availability
Source data underlying Figs. 1–4, Figs. 6–9 and Supplementary Figs. 6 and 7 are available as a Source Data file. All other data that support the findings of this study, the custom software used and the newly generated mouse lines will be made available upon request.

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

## Acknowledgements

We thank J. Robinson, T. Schuck, R. Gathagan, H. Brown, M. Olufemi, E. Meymand, C. Kim, H. Kim for technical assistance. We thank M. Hawk, B. Dombroski, and T. Bernard-Banks for 6hTau mouse breeding. We thank P. Davies at Hofstra Northwell School of Medicine for contributing PHF1 antibodies; J.A. Gonzalez at Massachusetts General Hospital for providing brain tissue from a PSP case. This work was funded by AG10124 (JQT), AG17586 (VMYL), CurePSP (JQT), Woods Foundation (VMYL), U54 NS100693 (GDS), and BrightFocus A2018802S (ZH).

## Author contributions

Z.H. designed the studies and generated most of the data along with J.D.M. and M.K. and interpreted all the results. J.L.G. and L.C. involved in the extraction of human brain lysates for injection. H.X. performed primary neuron culture experiments. S.N. provided the data of WT mice injected with AD, PSP, and CBD tau strains. B.Z. did mouse brain injection. S.K. assistant the surgeries and maintained T44mTauKO mice. G.G. generated the GT38 antibody. G.D.S. generated and bred the 6hTau mice. J.Q.T. participated in human tauopathy case studies, as well as in writing of the manuscript. Z.H. and V.M.L. supervised the study and wrote the manuscript, while all co-authors read and approved the manuscript it.

## Competing interests

The authors declare no competing interests.
