## [Peer Review File · Nature Communications]

REVIEWERS' COMMENTS:

Reviewer #2 (Remarks to the Author):

In this manuscript, Dr. He and Dr. Lee have examined the relative transmission efficacies of different tauopathy-derived brain lysates in a mouse model that expresses human 3R+4R isoforms. Using the 6h tau mice that they developed, the authors show that tau transmission does not depend on the tau isoforms but are influenced by the type of tauopathy that the seeds are derived from (i.e., tau seeds have 'unique pathological conformations').

This manuscript presents the data in extreme and painstaking details. First, the authors collected brain lysates (sarkosyl fraction) from AD, PSP, CBD and PiD patients. The 6h tau mice were injected into the hippocampus with these tau seeds (P0) and then these seeds were used to inject 6h tau mice (P1) and this was repeated to obtain the P2 generation. The idea is that if these tau seeds have prion properties, then serial passaging will magnify their pathologic properties, if you will. Surprisingly, the seeds lose their potency $P2 < P1 < P0$ - the question here arises regarding the prion like transmissibility of tau seeds, which is the basis of this study. Of note, Fig 2 h, I and j have different scales – if you fix the scales, this would be very apparent. Nevertheless, this study is important as it gives us a sense of how this seeding interacts with the presence of 3R and 4R tau. This is exemplified subsequently when they inject the tau seeds into a 4R tau mice (WT mice) and a 3R tau mice (T44mTauKO mice). For example, PiD does not involve 4R whereas PSP does not involve 3R isoform. In the next experiment, the authors generated 'purer' versions of 3R or 4R tau seeds by in vivo propagation and showed that pure tau isoforms derived from different tauopathies recruited the acceptor tau isoforms in very unique ways to form pathological tau. It is noteworthy that the authors selected to purify the seeds via in vivo propagation and not, say, IP techniques. The authors also show that while all of the tau seeds generate neuronal tauopathy, PSP-tau and CBD-tau preferentially result in non-neuronal tauopathy. Interestingly, from Fig 5, it seems that if you add the values of AT8+ staining from all cells of 6 month old CBD-tau injected mice, the total value is actually more than what the AT8+ data from AD-tau injected animals. This does not agree with Fig. 2 western blot data. Overall, this study is important in two respects - 1) it establishes that AD-, PSP-, CBD- and PiD-tau seeds specifically recruit different tau isoforms in transgenic mice and 2) these tau seeds generate cell-type specific pathologies in transgenic mice.

Some issues that need attention are:

- 1) some biophysical details on the unique pathological conformations inherent in tau derived from the different tauopathies will be great. The authors have provided non-native western blots and antibody immunoreactivity but these assays hardly can be used to define 'conformations'.
- 2) The authors are requested to include a neuroanatomic connectome showing how these different disease-derived seeds are transmitted anterogradely and retrogradely from the point of injection.
- 3) Could the authors comment why PiD and CBD-derived lysates perform so poorly? Is this because the pathologic tau seeds for these diseases reside in a non-sarkosyl fraction? Have the authors tried injecting soluble tau seeds from these human brains?
- 4) Could the authors comment on the fact that WT mice and T44mTauKO are not comparable? Especially because the T44 mice express humanized tau approximately 5x higher than endogenous mice tau (MGI:2385580)? Given that transgenic mice are 'better' in seeding, then could their data be a reflection of the different amounts of expressed tau in WT vs T44 mice?
- 5) The authors could explain why they did not use IP to 'purify' the 3R or 4R seeds for cross isoform-seeding experiments.
- 6) The authors need to address the discrepancy between Fig 2f and Fig 5 (e, f, g) 6 month timepoint – total AT8 immunoreactivity between AD-tau and CBD-tau groups.

7) It is very surprising that all tau seeds resulted in silver positivity and MC1 (Suppl. Fig. 4) whereas only AD-tau and PSP-tau seeds predominantly showed ThioS (Fig. 7). Can the authors comment on this?

8) Please provide individual datapoints in the bar graphs in Fig 2 (as has been done in the rest of the manuscript).

Reviewer #3 (Remarks to the Author):

Thanks for giving me the opportunity to review the manuscript by He et al., titled "Transmission of Tauopathy Strains is Independent of Their Isoform Composition". The revised manuscript addressed most of the questions raised by the reviewers and is much improved compared with previous submission. The revised manuscript shed new insight on the mechanism of pathogenesis of distinct tauopathies, provide important new evidence about the strain difference in tauopathies, which implicates that conformations but not the isoform compositions as the key intrinsic properties of different tau strains. one main concern is that one critical term "Propagation" was used multiple times in the manuscript to describe the tau passage in the mouse models. However, the meaning "propagation" was not well defined and may cause certain confusion to the field. It was not clear to me whether "propagation" in the context means pathology amplification of tau induced by the inoculation of insoluble tau extract inside a cell, or the pathological spreading of tau pathology from one cells to other cells? While the data presented in the manuscript would support the notion that inoculation of insoluble tau species from human patients or mouse brains can induce accumulation of the phosphorylated and certain conformational changes of tau in the cells, there is not enough evidence to support the view that the tau pathology is capable of propagating to other cells in this study.

A key observation In Figure 2 is that over the generations there is a dilution or decrease of the pathology. The authors explained that the decrease in the induced pathologies shown in Fig. 2 was due to the differences in the amount of seeds used, but not in the decrease in seeding potency. This new supporting data was included in Fig. 6 of the revised manuscript. The data showed in Fig 6 suggested that P1 tau pathologies were capable of inducing similar pathologies at the same dose of P0 tau, in terms of cell type specificity and amount. However, it could not explained why the tau pathology in P1 mice could not be amplified to the level of P0. This data also did not prove that P0 or P1 tau pathology can induced the in vivo propagation in a prion-like manner.

Prions are infectious proteins that capable of induce the conformation converting the prion protein to the aberrant form through a process of nucleated polymerization and amplification. A prion-like propagation would predict the pathology will be amplified and spreading to wider area. However, data presented in Fig 2 did not show clear evidence of amplification of tau pathology over the generations. Furthermore, data presented in Fig3c and 3d, Fig 7a and 7b, supplementary Fig 3, and supplementary Fig 6, showed that the level and distribution of tau accumulation were similar at 3mpi and 6mpi, suggesting that no significant amplification and propagation of tau occurred between 3 to 6 month, which is not consistent with the notion that P0 or P1 tau pathology can induced the in vivo propagation in a prion-like manner.

I agree with the author's hypothesis that initial tau pathogenesis could be divided into two phases: the initial misfolded seed formation phase followed by amyloidogenic amplification phase, where the endogenous monomeric tau isoforms are templated by misfolded seeds to adopt misfolding conformations followed by transformation into pathological forms. However, this hypothesis did not address the issue of whether initial tau pathogenesis will gain the ability to spreading and propagate to other cells. The authors also agreed that the pathological tau may exist as multiple stages, but they were uncertain which stage of tau pathology is correlated with the propagate ability of this pathology. I would strongly suggest the author to define staging of tau progression, and propagation of tau more clearly in the manuscript.

Some minor issues:

In the rebuttal the authors said that "biochemical characterization of P1 and P2 also showed those induced pathologies were sarkosyl-insoluble, a key feature for pathological tau proteins." I agree sarkosyl-insoluble is a key feature for pathological tau proteins. But we also should be aware multiple stages of tau pathology exist in tauopathy. sarkosyl-insoluble fractions of tau only contains tau tangles, but also pretangle aggregates (Greenberg SG, Davies P. 1990.)

In the discussion, they stated "we generated a new tauopathy mouse model to study the intrinsic properties of Htau strains..." which is not accurate. The 6htau is a mouse model mouse line expressing both 3R and 4R Htau isoforms, but not a model of tauopathy.

Response to Reviewers' comments:

Response to Reviewer #2 (Remarks to the Author):

“...The idea is that if these tau seeds have prion properties, then serial passaging will magnify their pathologic properties, if you will. Surprisingly, the seeds lose their potency $P2 < P1 < P0$ - the question here arises regarding the prion like transmissibility of tau seeds, which is the basis of this study. Of note, Fig 2 h, I and j have different scales – if you fix the scales, this would be very apparent.”

There is a misunderstanding here. As we clarified in our previous response to Reviewer 2, the differences observed in induced pathology for P0, P1 and P2 were largely due to the different amount of the seeds injected into the mice. To avoid such misunderstanding, we stated in the main text (Line 175) that “*Due to the limited yield of strain_P1, P2 from the injected mice, we could not inject as much propagated pathological tau as P0 into the 6hTau mice in this in vivo propagation experiment, resulting in lower pathologies induced by strain_P1 and strain_P2 when compared with P0*”. To further clarify differences in the dosage, we provided detailed dosage information in Fig. 2 figure legend and we state that: “*Due to the low recovery, the P1 injection doses were 0.1-0.5 μg per site, and P2 were 0.01-0.03 μg per site, while the P0 were 1 μg per site.*”. Since there were 10 fold differences in the injected dosages among each generation, it would not be surprising that the induced pathologies by P0, P1 and P2 were quite different as shown in Fig. 2b, d, e, h, i and j.

Furthermore, we rigorously compared the potency differences between P0 and P1 for each strains, using both cell culture (*in vitro*) and animal (*in vivo*) assays. As shown in Fig. 8 (original Fig. 6), it is clear that the potency of P1 was not decreased compared to P0 for each strain in both assays.

Some issues that need attention are:

1) some biophysical details on the unique pathological conformations inherent in tau derived from the different tauopathies will be great. The authors have provided non-native western blots and antibody immunoreactivity but these assays hardly can be used to define ‘conformations’.

This is a valid point but this reviewer already felt that “*The manuscript presents the data in extreme and painstaking detail*”. Adding more biophysical details on the unique pathological conformations for each tau strain derived from the different tauopathies may provide additional

insights but it also will further increase the amount of data in the manuscript. Thus, we believe the proposed biophysical studies are beyond the scope of our current paper and would be better to include in another separate study.

Although we did not provide direct biophysical details on the different strains in this manuscript, we did provide other evidence to support unique pathological conformations by using tau conformational specific antibodies (i.e. MC1, GT38) for immunostaining and by demonstrating that Thio S, an amyloid structure specific dye, binds indirectly to the distinct conformations of different strains. Furthermore, in our previous paper, we used partial proteinase K digestion to demonstrate the potential conformational differences among the different strains (please see Fig. 1F-H in Narasimhan et al., J. Neurosci, 2017). We cited this paper on line 346 in the revised manuscript. Using these different approaches, we believe we have convincingly showed the presence of distinct tau strain conformation in the manuscript. Currently, we are collaborating with structure biologists to elucidate the biophysical properties of the distinct tau strain conformations.

2) The authors are requested to include a neuroanatomic connectome showing how these different disease-derived seeds are transmitted anterogradely and retrogradely from the point of injection.

As suggested, we have added in Fig. 5a (original Fig. S6) a neuroanatomic connectome map in the revised manuscript, showing the anterograde and retrograde connections between the injection site (Dorsal hippocampus) and other major brain regions. Combined with the pathology heatmaps of each strain shown in Fig. 5b-e, we believe that we have provided the readers the information about how these different strains of tau seeds are transmitted from the site of injection *in vivo*.

3) Could the authors comment why PiD and CBD-derived lysates perform so poorly? Is this because the pathologic tau seeds for these diseases reside in a non-sarkosyl fraction? Have the authors tried injecting soluble tau seeds from these human brains?

This point is well taken. Since all different strains could induce neuronal pathologies, here we use the seeding ability of neuronal tau pathologies as the standard for our readout to discuss this point. As shown in Fig. 6e (original Fig. 5), in terms of neuronal pathologies, the

potencies of PiD-tau and CBD-tau were low before 3 months post injection, but their potency caught up by 6 months post injection, and were comparable to that of AD-tau and PSP-tau, indicating it is not the absolute potency, but the seeding kinetics that are different among distinct strains. One possibility is that the PiD-tau and CBD-tau seeds are slower in the initial seeding stage, but once initiated, the elongation phase becomes fast. Alternatively, degradation of the different misfolding proteins could be different since it is possibly the seeded pathologies by PiD-tau and CBD-tau are more easily degraded initially, but over time, the protein degradation systems are overwhelmed by the accumulated misfolding proteins resulting in more pathology at later time points.

This reviewer also brought up an important and interesting point that if active soluble tau seeds are present in the sarkosyl-soluble fraction. It is possible the solubility of distinct strain seeds are different since they have distinct conformations, and it would be interesting to inject the soluble tau derived from different tauopathy brains and compare their potency *in vivo*, which we have not done yet. However, we did examine this question in our *in vitro* cell culture assay using the sarkosyl-soluble fractions from AD, CBD and PSP brains, and compared their activities to our final enriched insoluble tau fractions from each tauopathy. Unfortunately, the sarkosyl-soluble fractions were too toxic to the neurons, likely due to the presence of 1% sarkosyl (data not shown). Instead, we devised a slightly modified version of our extraction protocol: we homogenized brain tissue in the same high-salt buffer as in our standard enrichment protocol (see Appendix Fig. 1A below), then either took the supernatant in the same buffer (sup 2) or dialyzed the supernatant and re-suspended in PBS (sup 3). We then tested the seeding activities of sup 2 and sup 3 in CD1 neurons, but we found that sup 2 was still toxic to neurons likely due to the presence of 0.1% sarkosyl (data not shown). Sup 3 at the same concentration of tau as the enriched insoluble tau for each lysate (200 ng tau per coverslip) from AD, CBD, and PSP had little to no activity in CD1 neurons compared to the high activity of the final insoluble tau fraction (Appendix Fig. 1C). Finally, no tau pathology was detected in neurons treated with any of the three soluble tau fractions as well as the insoluble fraction from a control brain (Appendix Fig. 1C). Therefore, we conclude that our soluble tau fractions obtained from AD, CBD and PSP brains contained very little, if any, active seeds, and that most of the pathological, active tau remains in the enriched insoluble fractions.

4) Could the authors comment on the fact that WT mice and T44mTauKO are not comparable? Especially because the T44 mice express humanized tau approximately 5x higher than endogenous mice tau (MGI:2385580)? Given that transgenic mice are 'better' in seeding, then could their data be a reflection of the different amounts of expressed tau in WT vs T44 mice?

Our understanding of the reviewer's concern here is that since WT mice express much less tau than T44 mice, the absence of pathologies in PiD-tau injected WT (PiD-WT) mice may be due to the lower amount of tau expression in WT mice that are insufficient for recruitment by strain PiD-tau, rather than our interpretation in the manuscript that 3R tau strain is inefficient to seed 4R tau monomers.

We think it is unlikely that the lack of pathologies in PiD-WT mice was due to too little tau expression in WT mice. As shown in Fig. 3e, f, we examined tau pathology in these WT mice following 3 and 6 months post injection. By 6 months post-injection, most other tau strains would induce decent amount of tau pathologies, but there was still almost no pathologies induced by PiD-tau, suggesting that the inability of PiD-tau to induce pathologies in WT mice was not due to tau expression levels. Furthermore, in the 6hTau mice, which have higher total tau expression than WT mice (Fig. 1b), the PiD-tau (Fig. 2 h), and purified PiD-3R-tau (Fig. 4f) are still inefficient in recruiting 4R tau monomers, supporting our claim that 3R tau strain is inefficient to seed 4R tau monomers.

Conversely, although total tau expression in WT (4R Tau) is less than in T44mTauKO (3R tau) mice (Fig. 1b), the same dose of PSP-, CBD-tau still induced abundant pathologies in WT mice, but only negligible levels in T44mTauKO mice (Fig. 3e, f). These and other data provided here consistently demonstrate that the different templating abilities were not due to the monomer expression level but more likely related to the unique properties of each tau strain.

5) The authors could explain why they did not use IP to 'purify' the 3R or 4R seeds for cross isoform-seeding experiments.

In the study we aimed to convert AD strain isoform composition from a mixture 3R/4R into a pure 3R or 4R strain rather than to purify the 3R or 4R seeds from AD brains. Furthermore, based on our study (Guo et al., JEM, 2016) and others (Fitzpatrick et al., Nature, 2017), the interprotofilament packing of AD-tau fibrils suggest that they are comprised of both 3R and 4R tau

monomers, and as a result it is impossible to separate AD-tau to obtain pure 3R and 4R seeds. Finally, we are not aware of any high affinity 3R or 4R isoform-specific antibodies suitable for quantitative and efficient IP experiments.

6) The authors need to address the discrepancy between Fig 2f and Fig 5 (e, f, g) 6 month timepoint – total AT8 immunoreactivity between AD-tau and CBD-tau groups.

Fig. 2f could not be directly compared with Fig. 6e-g (original Fig. 5), since the measurements were done in different ways, and the time scales were also different.

In Fig. 2f, the induced tau pathologies were measured by the amount of total insoluble tau protein using western blots. In Fig. 6e-g, the induced tau pathologies were measured by the number of different cell types that are harboring pathological tau. The total amount of pathological tau protein could not be a simple correlation with the total number of AT8+ cells, since there are variable compositions of the total number of neurons and glial cells. Considering the relative size of neurons and glial cells, the total tau pathologies in one single neuron would be much more than those in a single oligodendrocyte or astrocyte, as indicated in Fig. 6 b, c and d (please compare the area of red fluorescence in a neuron and a glial cell).

Furthermore, the data in Fig. 2f were generated at 3 months post injection, while the data in Fig. 6e-g were generated from 6 months post injection. Finally, the seeding kinetics for AD-tau and CBD-tau are different, so these two data sets could not be compared directly per the reviewer.

7) It is very surprising that all tau seeds resulted in silver positivity and MC1 (Suppl. Fig. 4) whereas only AD-tau and PSP-tau seeds predominantly showed ThioS (Fig. 7). Can the authors comment on this?

ThioS dye binds specifically to β -pleated sheet amyloid structures but silver staining is less specific and binds to filamentous structures including neurofilaments. The differences in specificities may explain the different staining patterns between Fig. 9 a, b (original Fig. 7) and Fig. S4. Furthermore, the data from these experiments were presented differently. Fig. 9 shows the percentage of ThioS staining among the different strain-induced pathologies, whereas Fig. S4 shows the existence of silver-positive pathologies in each group. Although there are a lot of AT8+ pathologies, the silver staining was generally very limited, and was only detectable in

entorhinal cortex region in most cases. In the manuscript, we only stated that the pathologies induced by AD-, PSP- and CBD-, but not PiD-tau were positive in silver (Line 142).

8) Please provide individual datapoints in the bar graphs in Fig 2 (as has been done in the rest of the manuscript).

As suggested, we replaced the Fig. 1d, h, I, k, m, n; Fig. 2 h, i, j; Fig. 3f and Fig. 7 b, c, e, f and h (original Supplementary Fig. S7) into scatter plots in the revised manuscript.

Response to Reviewer #3 (Remarks to the Author):

“...one main concern is that one critical term “Propagation” was used multiple times in the manuscript to describe the tau passage in the mouse models. However, the meaning “propagation” was not well defined and may cause certain confusion to the field.

It was not clear to me whether “propagation” in the context means pathology amplification of tau induced by the inoculation of insoluble tau extract inside a cell, or the pathological spreading of tau pathology from one cells to other cells? While the data presented in the manuscript would support the notion that inoculation of insoluble tau species from human patients or mouse brains can induce accumulation of the phosphorylated and certain conformational changes of tau in the cells, there is not enough evidence to support the view that the tau pathology is capable of propagating to other cells in this study.

A key observation In Figure 2 is that over the generations there is a dilution or decrease of the pathology. The authors explained that the decrease in the induced pathologies shown in Fig. 2 was due to the differences in the amount of seeds used, but not in the decrease in seeding potency. This new supporting data was included in Fig. 6 of the revised manuscript. The data showed in Fig 6 suggested that P1 tau pathologies were capable of inducing similar pathologies at the same dose of P0 tau, in terms of cell type specificity and amount. However, it could not explained why the tau pathology in P1 mice could not be amplified to the level of P0. This data also did not prove that P0 or P1 tau pathology can induced the in vivo propagation in a prion-like manner.

Prions are infectious proteins that capable of induce the conformation converting the prion protein to the aberrant form through a process of nucleated polymerization and amplification. A prion-like propagation would predict the pathology will be amplified and spreading to wider area. However, data presented in Fig 2 did not show clear evidence of amplification of tau pathology over the generations. Furthermore, data presented in Fig3c and 3d, Fig 7a and 7b, supplementary Fig 3, and supplementary Fig 6, showed that the level and distribution of tau accumulation were similar at 3mpi and 6mpi, suggesting that no significant amplification and propagation of tau occurred between 3 to 6 month, which is not consistent with the notion that P0 or P1 tau pathology can induced the *in vivo* propagation in a prion-like manner.

I agree with the author's hypothesis that initial tau pathogenesis could be divided into two phases: the initial misfolded seed formation phase followed by amyloidogenic amplification phase, where the endogenous monomeric tau isoforms are templated by misfolded seeds to adopt misfolding conformations followed by transformation into pathological forms. However, this hypothesis did not address the issue of whether initial tau pathogenesis will gain the ability to spreading and propagate to other cells. The authors also agreed that the pathological tau may exist as multiple stages, but they were uncertain which stage of tau pathology is correlated with the propagate ability of this pathology. I would strongly suggest the author to define staging of tau progression, and propagation of tau more clearly in the manuscript."

Reviewer 3's point that we need to define the term "propagation" to avoid confusion in the field is valid. In the revised manuscript Line 73, we provided our conceptual definition for the term of "propagation" that "pathological tau protein could transmit their pathological conformations to the physiological tau protein, converting endogenous tau protein from its normal forms into pathological forms." We further provided the technical definition on how we conducted propagation in Line 129 that "we propagated the tau strains *in vivo* by intracerebrally injecting ...different human brain-derived tau strains ... into the hippocampus and overlying cortex of ... mice". With these edits, we hope we could clarify to the reviewers and the readers what we mean by the term "propagation".

In terms of prion-like issue raised again by Reviewer 3, whether pathological tau is a prion or not is still hotly debated in the field, but it is not the key point we want to convey to the readers in this manuscript. To avoid such misunderstanding, we replaced the word "prion-like"

with the word “unique” in Line 72 in the revised manuscript. Whether pathological proteins can be amplified during propagation are affected by multiple factors, such as the seed properties (or seeding kinetics), incubation time post-injection, methods of amplification and so on. For example, when we consider the incubation time, we have to consider the differences between time scale in year that pathological seeds propagate in human brain versus the time scale in month that pathological seeds propagate in experimental mouse models. Furthermore, considering the unknown percentage of the injected seeds that enter the cells and effective as seeds (please see Supplementary Fig. 5), and the percentage yield of all the induced pathologies *in vivo*, it is hard to provide an accurate estimation of the actual amplification rate. As a result, we could not draw the same conclusion as Reviewer 3 from the current data in the manuscript that the pathologies were not amplified during the *in vivo* propagation.

In terms of defining the staging of tau progression, since we still not fully understand the pathological tau propagation process and the underlying mechanisms, rather than specifically defining a specific time point for a certain stage, we could only provide a working hypothesis for pathological tau propagation.

Some minor issues:

In the rebuttal the authors said that “biochemical characterization of P1 and P2 also showed those induced pathologies were sarkosyl-insoluble, a key feature for pathological tau proteins.” I agree sarkosyl-insoluble is a key feature for pathological tau proteins. But we also should be aware multiple stages of tau pathology exist in tauopathy. sarkosyl-insoluble fractions of tau only contains tau tangles, but also pretangle aggregates (Greenberg SG, Davies P. 1990.)

The Reviewer’s point here is well taken. Insoluble tau is generally considered to contain pathological tau, but pathological tau may also exist as other forms, and could be present in other fractions at multiple stages.

In the discussion, they stated “we generated a new tauopathy mouse model to study the intrinsic properties of Htau strains...” which is not accurate. The 6htau is a mouse model mouse line expressing both 3R and 4R Htau isoforms, but not a model of tauopathy.

We thank the Reviewer 3 in pointing out a potential misleading statement in the manuscript. Here, we have clarified “a new tauopathy mouse model” that “by inoculating distinct tauopathy brain-derived pathological tau strains into a new mouse line 6hTau that express equal ratio of 3R and 4R tau isoforms, ...” in **Line 353** in the revised manuscript.

References:

Narasimhan Sneha, Guo L. Jing, Changolkar Lakshmi, Stieber Anna, McBride D. Jennifer, Silva Luisa, He Zhuohao, Zhang Bin, Gathagan J. Ronald, Trojanowski Q. John, and Lee M.-Y. Virginia: Pathological tau strains from human brains recapitulate the diversity of tauopathies in non-transgenic mouse brain. *The Journal of neuroscience*: 37(47): 11406-11423, Nov 2017.

Guo L. Jing, Narasimhan Sneha, Changolkar Lakshmi, He Zhuohao, Stieber Anna, Zhang Bin, Gathagan Ronald J, Iba Michiyo, McBride Jennifer D, Trojanowski John Q, Lee Virginia M Y: Unique pathological tau conformers from Alzheimer's brains transmit tau pathology in nontransgenic mice. *The Journal of experimental medicine* 213(12): 2635-2654, Nov 2016.

Fitzpatrick W. Anthony, Falcon Benjamin, He Shaoda, Murzin G. Alexey, Murshudov Garib, Garringer J. Holly, Crowther R. Anthony, Ghetti Bernardino, Goedert Michel and Sheres H. Sjors: Cryo-EM structures of tau filaments from Alzheimer's disease 547(7662): 185-190, Jul 2017.

Appendix Figure 1: **A**, Modified sequential extraction protocol to isolate soluble tau from AD, CBD, and PSP brains without sarkosyl. **B**, Western blot using pan-tau antibody 17025 (1:1000 red) and phosphorylated tau antibody PHF-1 (1:1000 green) for homogenate (1), supernatant in PHF buffer (2), and supernatant in PBS (3) from AD, CBD, and PSP brain extractions. Western blot shows tau in all three fractions, but phosphorylated tau is found in the homogenate and not the supernatant fractions, consistent with phosphorylated tau being insoluble and being extracted in the pellet fraction. **C**, Seeding of primary CD1 neurons with PBS soluble fractions of AD, CBD, and PSP compared to enriched insoluble tau (AD-tau, CBD-tau, and PSP-tau) at 200 ng of tau per coverslip (as

described in our methods). After 15 days' seeding, immunocytochemistry for mouse tau specific antibody T49 (1:1000) shows little to no activity of the soluble fractions compared to high activity of AD-tau, CBD-tau, and PSP-tau. Control brain insoluble fraction was tested as a negative control, which also shows no seeding activity as shown in Fig. 2c in the manuscript. Scale bar: 100 μ m. Quantification of T49 signal/DAPI counts (tau pathology/cell) shows much higher seeding activity of insoluble tau fractions compared to soluble tau from AD, CBD, and PSP brains.